# No thick carbon dioxide atmosphere on the rocky exoplanet TRAPPIST-1 c

Sebastian Zieba[1,2 ✉], Laura Kreidberg[1], Elsa Ducrot[3], Michaël Gillon[4], Caroline Morley[5], Laura Schaefer[6], Patrick Tamburo[7,8], Daniel D. B. Koll[9], Xintong Lyu[9], Lorena Acuña[1,10], Eric Agol[11,12], Aishwarya R. Iyer[13], Renyu Hu[14,15], Andrew P. Lincowski[11,12], Victoria S. Meadows[11,12], Franck Selsis[16], Emeline Bolmont[17,18], Avi M. Mandell[19,20] & Gabrielle Suissa[11,12]

Seven rocky planets orbit the nearby dwarf star TRAPPIST-1, providing a unique opportunity to search for atmospheres on small planets outside the Solar System[1]. Thanks to the recent launch of the James Webb Space Telescope (JWST), possible atmospheric constituents such as carbon dioxide ($CO_2$) are now detectable[2,3]. Recent JWST observations of the innermost planet TRAPPIST-1 b showed that it is most probably a bare rock without any $CO_2$ in its atmosphere[4]. Here we report the detection of thermal emission from the dayside of TRAPPIST-1 c with the Mid-Infrared Instrument (MIRI) on JWST at 15 μm. We measure a planet-to-star flux ratio of $f_p/f_* = 421 \pm 94$ parts per million (ppm), which corresponds to an inferred dayside brightness temperature of $380 \pm 31$ K. This high dayside temperature disfavours a thick, $CO_2$-rich atmosphere on the planet. The data rule out cloud-free $O_2/CO_2$ mixtures with surface pressures ranging from 10 bar (with 10 ppm $CO_2$) to 0.1 bar (pure $CO_2$). A Venus-analogue atmosphere with sulfuric acid clouds is also disfavoured at $2.6\sigma$ confidence. Thinner atmospheres or bare-rock surfaces are consistent with our measured planet-to-star flux ratio. The absence of a thick, $CO_2$-rich atmosphere on TRAPPIST-1 c suggests a relatively volatile-poor formation history, with less than $9.5^{+7.5}_{-2.3}$ Earth oceans of water. If all planets in the system formed in the same way, this would indicate a limited reservoir of volatiles for the potentially habitable planets in the system.

Little is known about the compositions of terrestrial exoplanet atmospheres, or even whether atmospheres are present at all. The atmospheric composition depends on many unknown factors, including the initial inventory of volatiles, outgassing resulting from volcanism and possible atmospheric escape and collapse (see, for example, ref. 5). Atmospheric escape may also depend on the spectral type of the host star: planets around M dwarfs may be particularly vulnerable to atmospheric loss during the long pre-main sequence phase[6]. The only way to robustly determine whether a terrestrial exoplanet has an atmosphere is to study it directly, through its thermal emission, reflected light or transmission spectrum. The tightest constraints on atmospheric properties so far have come from observations of the thermal emission of LHS 3844 b, GJ 1252 b and TRAPPIST-1 b. The measurements revealed dayside temperatures consistent with no redistribution of heat on the planet and no atmospheric absorption from carbon dioxide[4,7,8]. These results motivate observations of cooler planets, which may be more likely to retain atmospheres.

We observed four eclipses of TRAPPIST-1 c with MIRI on JWST in imaging mode. The observations took place on 27 October, 30 October, 6 November and 30 November 2022 as part of General Observer programme 2304. Each visit had a duration of approximately 192 min, covering the 42-min eclipse duration of TRAPPIST-1 c as well as out-of-eclipse baseline to correct for instrumental systematic noise. The observations used the MIRI F1500W filter, a 3-μm-wide bandpass centred at 15 μm, which covers a strong absorption feature from $CO_2$. Across the four visits, we collected 1,190 integrations in total using the FULL subarray. See Methods for further details on the design of the observations.

We performed four independent reductions of the data using the publicly available Eureka! code[9] as well as several custom software pipelines. Each reduction extracted the light curve of TRAPPIST-1 using aperture photometry (see Methods and Extended Data Table 2). We then fitted the light curves with an eclipse model and a range of different parameterizations for the instrumental systematics, including

[1]Max-Planck-Institut für Astronomie, Heidelberg, Germany. [2]Leiden Observatory, Leiden University, Leiden, The Netherlands. [3]Université Paris-Saclay, Université Paris Cité, CEA, CNRS, AIM, Gif-sur-Yvette, France. [4]Astrobiology Research Unit, University of Liège, Liège, Belgium. [5]Department of Astronomy, University of Texas at Austin, Austin, TX, USA. [6]Department of Earth and Planetary Sciences, Stanford University, Stanford, CA, USA. [7]Department of Astronomy, Boston University, Boston, MA, USA. [8]The Institute for Astrophysical Research, Boston University, Boston, MA, USA. [9]Department of Atmospheric and Oceanic Sciences, Peking University, Beijing, People's Republic of China. [10]Aix-Marseille Université, CNRS, CNES, Institut Origines, LAM, Marseille, France. [11]Astrobiology Program, Department of Astronomy, University of Washington, Seattle, WA, USA. [12]NASA Nexus for Exoplanet System Science, Virtual Planetary Laboratory Team, University of Washington, Seattle, WA, USA. [13]School of Earth and Space Exploration, Arizona State University, Tempe, AZ, USA. [14]Jet Propulsion Laboratory, California Institute of Technology, Pasadena, CA, USA. [15]Division of Geological and Planetary Sciences, California Institute of Technology, Pasadena, CA, USA. [16]Laboratoire d'Astrophysique de Bordeaux, Université de Bordeaux, CNRS, B18N, Pessac, France. [17]Observatoire Astronomique de l'Université de Genève, Versoix, Switzerland. [18]Centre Vie dans l'Univers, Université de Genève, Geneva, Switzerland. [19]NASA Goddard Space Flight Center, Greenbelt, MD, USA. [20]Sellers Exoplanet Environments Collaboration, NASA Goddard Space Flight Center, Greenbelt, MD, USA. ✉e-mail: zieba@mpia.de

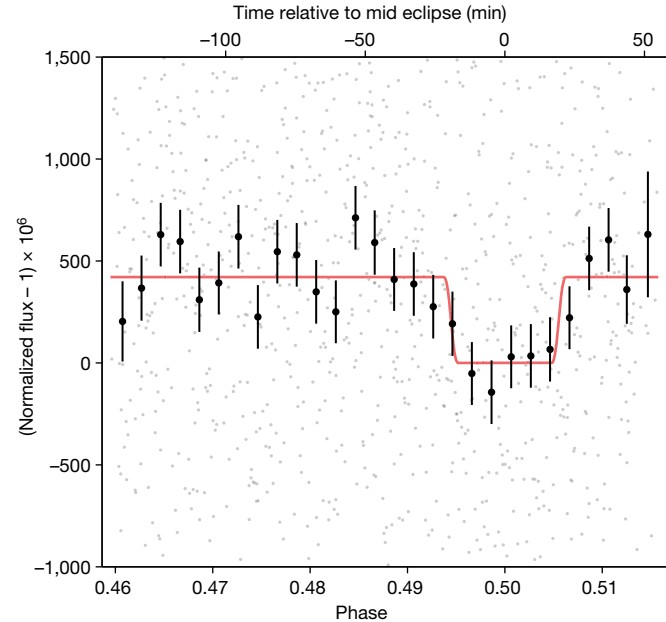

**Fig. 1 | Eclipse light curve of TRAPPIST-1 c taken with MIRI F1500W.** The phase-folded secondary eclipse light curve of TRAPPIST-1 c, measured with the MIRI imager at 15 μm. The eclipse is centred at orbital phase 0.5 and has a measured depth of $f_p/f_* = 421 \pm 94$ ppm. The light curve includes four visits (that is, four eclipses), each spanning approximately 3.2 h. To make the eclipse more easily visible, we binned the individual integrations (grey points) into 28 orbital phase bins (black points with 1σ error bars). The light curve was normalized and divided by the best-fit instrument systematic model. The best-fit eclipse model is shown with the solid red line. The data and fit presented in this figure are based on the SZ reduction, one of the four independent reductions we performed in this work.

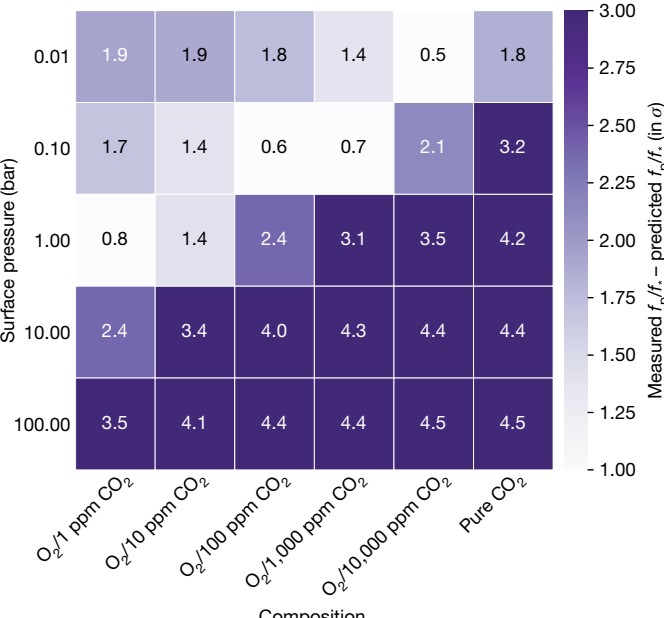

**Fig. 2 | Grid plot comparing a suite of atmospheric models to the measured eclipse depth.** Comparison between the measured eclipse depth and a suite of different $O_2/CO_2$, cloud-free atmospheres for TRAPPIST-1 c with varying surface pressures and compositions. Darker grid cells indicate that we more strongly rule out this specific atmospheric scenario. The number in each cell is the difference between each model and the observations in units of σ. The lower the modelled atmosphere is in the grid, the higher its surface pressure. The rightmost column shows pure $CO_2$ atmospheres. The other columns are $O_2$-dominated atmospheres with different amounts of $CO_2$, ranging from 1 ppm (=0.0001%) to 10,000 ppm (=1%).

a polynomial in time, exponential ramps and decorrelation against the position and width of the point spread function (PSF). For the different analyses, the scatter of the residuals in the fitted light curves had a root mean square (rms) variability ranging from 938 to 1,079 ppm, within 1.06–1.22 times the predicted photon noise limit when using a corrected gain value[10]. We estimated the eclipse depths using Markov chain Monte Carlo (MCMC) fits to the data, which marginalized over all the free parameters in the analysis. The resulting eclipse depths from the four data analyses are consistent and agree to well within 1σ (see Extended Data Table 3). The phase-folded light curve from one of the reductions can be seen in Fig. 1. To determine the final eclipse depth, we took the mean value and uncertainty from the different reductions. To account for systematic error owing to differences in data reduction and modelling choices, we also added a further 6 ppm to the uncertainty in quadrature, which corresponds to the standard deviation in the eclipse depth between the four analyses. The resulting eclipse depth is $f_p/f_* = 421 \pm 94$ ppm.

From the measured eclipse depth, we derive a brightness temperature of 380 ± 31 K for TRAPPIST-1 c. The innermost planet in the system, TRAPPIST-1 b, was found to have a brightness temperature of $503^{+26}_{-27}$ K (ref. 4). Compared with previous detections of thermal emission from small ($R_p < 2R_\oplus$) rocky planets (see Extended Data Fig. 1), these temperatures are more than 500 K cooler (the previous lowest measured brightness temperature was 1,040 ± 40 K for LHS 3844 b (ref. 7)). TRAPPIST-1 c is the first exoplanet with measured thermal emission that is comparable with the inner planets of the Solar System; Mercury and Venus have equilibrium temperatures of 440 K and 227 K, respectively, assuming uniform heat redistribution and taking the measured Bond albedo values ($A_{B,Mercury} = 0.068$, $A_{B,Venus} = 0.76$) from refs. 11,12. Our measured temperature for TRAPPIST-1 c is intermediate between the

two limiting cases for the atmospheric circulation for a zero-albedo planet: zero heat redistribution (430 K; expected for a fully absorptive bare rock) versus global heat redistribution (340 K; expected for a thick atmosphere). This intermediate value hints at either a moderate amount of heat redistribution by an atmosphere ($\varepsilon = 0.66^{+0.26}_{-0.33}$) or a non-zero Bond albedo for a rocky surface ($A_B = 0.57^{+0.12}_{-0.15}$) (following the parameterization described in ref. 13).

To further explore which possible atmospheres are consistent with the data, we compared the dayside flux with a grid of cloud-free, $O_2$-dominated models with a range of surface pressures (0.01–100.0 bar) and $CO_2$ contents (1–10,000 ppm). Also, we generated cloud-free, pure $CO_2$ atmospheres using the same surface pressures. The models account for both atmospheric heat redistribution and absorption by constituent gasses[2,7,14] and assume a Bond albedo of 0.1 (see Methods). $O_2/CO_2$ mixtures are expected for hot rocky planets orbiting late M-type stars as the planet's $H_2O$ photodissociates and escapes over time, leaving a desiccated atmosphere dominated by $O_2$ (refs. 6,15,16). Substantial $CO_2$ (up to about 100 bar) is expected to accumulate from outgassing and does not escape as easily as $H_2O$ (refs. 17,18). For these mixtures, the predicted eclipse depth decreases with increasing surface pressure and with increasing $CO_2$ abundance, owing to the strong $CO_2$ absorption feature centred at 15 μm. Strong inversions for a planet in this parameter space are not expected[19]. With our measured eclipse depth, we rule out all thick atmospheres with surface pressures $P_{surf} \geq 100$ bar (see Fig. 2). For the conservative assumption that the $CO_2$ content is at least 10 ppm, we rule out $P_{surf} \geq 10$ bar. For cloud-free, pure $CO_2$ atmospheres, we can rule out surface pressures $P_{surf} \geq 0.1$ bar. As the TRAPPIST-1 planets have precisely measured densities, interior-structure models can give constraints on the atmospheric surface pressures, that is, higher surface pressures would decrease the observed bulk density of the planet. Our findings here agree with these models,

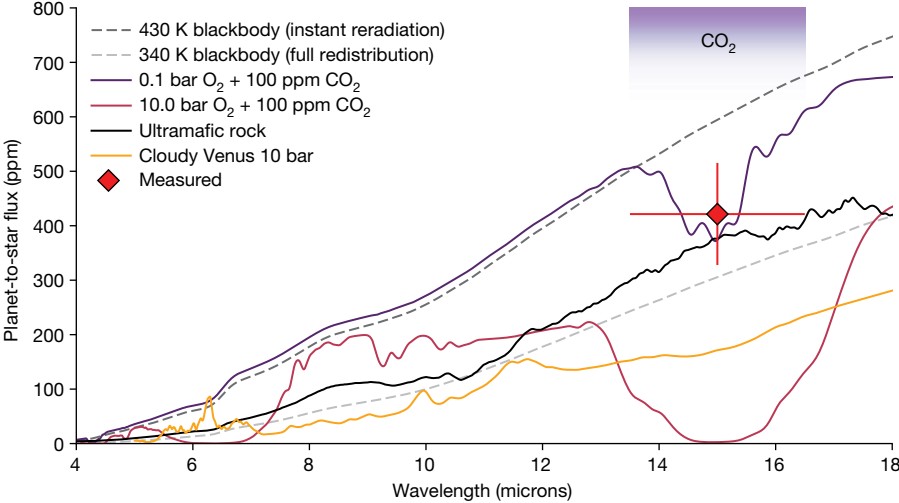

**Fig. 3 | Observed flux of TRAPPIST-1 c and various emission models.**
Simulated emission spectra compared with the measured eclipse depth of TRAPPIST-1 c (red diamond, with the vertical error bar representing the $1\sigma$ uncertainty on the measured eclipse depth). The $CO_2$ feature overlaps directly with the MIRI F1500W filter used for these observations. The two limiting cases for the atmospheric circulation for a zero-albedo planet (zero heat redistribution, that is, instant reradiation of incoming flux and global heat redistribution) are marked with dashed lines. Two cloud-free, $O_2/CO_2$ mixture atmospheres are shown with purple and red solid lines. They show decreased emission at 15 μm owing to $CO_2$ absorption. A bare-rock model assuming an unweathered ultramafic surface of the planet with a Bond albedo of 0.5 is shown by the solid black line (see text for more information on weathering, including a full comparison of our measurement to a suite of surfaces in Extended Data Fig. 5). The cloudy Venus forward model with a surface pressure of 10 bar is shown with a solid yellow line.

which put an upper limit of 160 bar (80 bar) on the surface pressure at a $3\sigma$ ($1\sigma$) level[20].

We also compared the measured dayside brightness with several physically motivated forward models inspired by Venus. The insolation of TRAPPIST-1 c is just 8% greater than that of Venus[21], so it is possible that the two planets could have similar atmospheric chemistry. We

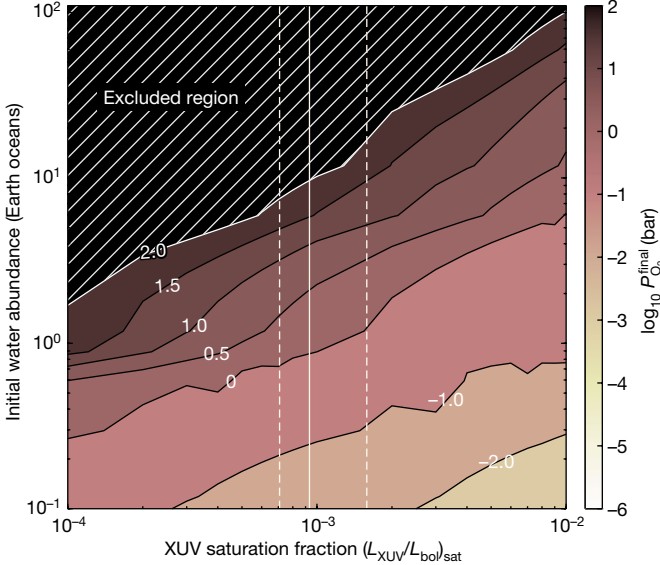

**Fig. 4 | Final oxygen atmospheric pressure for TRAPPIST-1 c after 7.5 Gyr of energy-limited escape.** We explore different initial planetary water abundances and the amount of XUV the planet receives during the star's saturated activity period[6], described as a fraction of its total bolometric luminosity. The vertical lines represent the nominal XUV saturation fraction of $\log_{10}(L_{XUV}/L_{bol}) = -3.03^{+0.23}_{-0.12}$ as estimated in ref. 25. We assume an escape efficiency of 0.1. The white numbers are the contour values for the logarithm of the atmospheric pressure in bar. Our upper limit on surface pressure of 10–100 bar implies an initial water abundance of approximately 4–10 Earth oceans.

used a coupled climate-photochemistry model to simulate an exact Venus-analogue composition (96.5% $CO_2$, 3.5% $N_2$ and Venus lower atmospheric trace gases), both with and without $H_2SO_4$ aerosols[3] (see Methods). The assumed surface pressure was 10 bar, which would produce similar results to a true 93-bar Venus analogue because, for both cases, the emitting layer and cloud deck lie at similar pressures. We find that these cloudy and cloud-free Venus-like atmospheres are disfavoured at $2.6\sigma$ and $3.0\sigma$, respectively (see Fig. 3 for the 10-bar cloudy Venus spectrum). The cloudy case is marginally more consistent with the data because the $SO_2$ aerosols locally warm the atmosphere, providing a warmer emission temperature within the core of the 15-μm band and therefore a larger secondary eclipse depth.

Finally, we compared the measured flux with bare-rock models with a variety of surface compositions, including basaltic, feldspathic, Fe-oxidized (50% nanophase haematite, 50% basalt), granitoid, metal-rich ($FeS_2$) and ultramafic compositions[22]. We also considered space weathering for these models, as TRAPPIST-1 c should have been substantially weathered owing to its proximity to the host star. On the Moon and Mercury, space weathering darkens the surface by means of the formation of iron nanoparticles[23]. On TRAPPIST-1 c, this process would similarly darken the surface and therefore increase the eclipse depth. We find that all bare-rock surfaces are consistent with the data (see Fig. 3 for an unweathered ultramafic surface and Extended Data Fig. 5 for all surfaces that we considered). Overall, fresh low-albedo surfaces (for example, basalt) or weathered surfaces are all compatible with the data, comparable with the probable bare-rock exoplanet LHS 3844 b (ref. 7). The highest albedo models, unweathered feldspathic and granitoid surfaces, are a marginally worse fit (consistent at the $2\sigma$ level).

To put our results into context with the formation history of the planet, we ran a grid of atmospheric evolution models over a range of initial water inventories (0.1–100 Earth oceans) and extreme ultraviolet (XUV) saturation fractions for the host star ($10^{-4}$–$10^{-2}$) (see Fig. 4). The model incorporates outgassing, escape of water vapour and oxygen and reaction of oxygen with the magma ocean[15]. For an XUV saturation fraction of $10^{-3}$ being a typical value for a low-mass star[24], we find

that the final surface pressure of oxygen could range over several orders of magnitude (0.1–100 bar), depending on the initial water inventory (see Fig. 4). Our measured eclipse depth disfavours surface pressures at the high end of this range (greater than 100 bar) for conservative $CO_2$ abundances, implying that TRAPPIST-1 c most probably formed with a relatively low initial water abundance of less than $9.5^{+7.5}_{-2.3}$ Earth oceans. For higher $CO_2$ abundances (>10 ppm), we rule out surface pressures greater than 10 bar, implying that the planet formed with less than $4.0^{+1.3}_{-0.8}$ Earth oceans. Our result suggests that rocky planets around M-dwarf stars may form with a smaller volatile inventory or experience more atmospheric loss than their counterparts around Sun-like stars. This finding motivates further study of the other planets in the TRAPPIST-1 system to assess whether a low volatile abundance is a typical outcome, particularly for the planets in the habitable zone.

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

## Methods

### JWST MIRI observations

As part of JWST General Observer programme 2304 (principal investigator: L. Kreidberg)[26], we observed four eclipses of the planet TRAPPIST-1 c (see Extended Data Table 1). They were taken on 25 October, 27 October, 30 October and 6 November 2022 with JWST's MIRI instrument using the F1500W filter. The observations used the FULL subarray with FASTR1 readout and 13 groups per integration. Each visit had a duration of approximately 3.2 h. We did not perform target acquisition for any of the visits because it was not enabled for MIRI imaging observations during cycle 1. However, the blind pointing precision of JWST was perfectly sufficient to place the target well centred on the field of view of the full array (74″ × 113″). Extended Data Fig. 2 shows one of the integrations with the FULL array.

### Data reduction

We performed four different reductions of the data collected for JWST General Observer programme 2304. The assumptions made by the reductions are listed in Extended Data Table 2. In the following, we describe the individual reductions.

**Data reduction SZ.** For our primary data reduction and data analysis, we used the open-source Python package Eureka![9], which is an end-to-end pipeline for time-series observations performed with JWST or the Hubble space telescope. We started our reduction with the raw uncalibrated ('uncal') FITS files, which we downloaded from the Mikulski Archive for Space Telescopes (MAST), and followed the multi-stage approach of Eureka! to generate a light curve for TRAPPIST-1 c. Eureka! has been previously successfully used to reduce and analyse the first JWST observations of exoplanets[27–31].

Stages 1 and 2 of Eureka! serve as a wrapper of the JWST Calibration Pipeline[32] (version 1.8.2). Stage 1 converts groups to slopes and applies basic detector-level corrections. We used the default settings for all steps in this stage but determined a custom ramp-jump detection threshold for each visit by minimizing the median absolute deviation of the final light curves. This step detects jumps in the up-the-ramp signal for each pixel by looking for outliers in each integration that might be caused by events such as cosmic rays. We determined a best jump detection threshold of $7\sigma$, $6\sigma$, $7\sigma$ and $5\sigma$ for visits 1, 2, 3 and 4, respectively, compared with the default value of $4\sigma$ set in the JWST pipeline. In stage 2, we only skipped the photom step to leave the data in units of DN $s^{-1}$ and not convert into absolute fluxes. In stage 3 of Eureka!, we first masked pixels in each visit that were flagged with a 'DO NOT USE' data quality entry, indicating bad pixels identified by the JWST pipeline. Next, we determined the centroid position of the star by fitting a 2D Gaussian to the source. JWST remained very stable during our observations of TRAPPIST-1 c and our target stayed well within a 0.01-pixel area (see Extended Data Table 1 and Extended Data Fig. 3). We recorded the centroid position in $x$ and $y$ and the width of the 2D Gaussian in $x$ and $y$ over time to be used in the fitting stage. Next, we determined the best target and background apertures by minimizing the rms of the final light curve. We therefore determined a target aperture of 4 pixels and a background annulus from 25 to 41 pixels from the centroid for each visit. The light curves show a ramp-like trend at the beginning of the observations, which has already been observed in previous JWST MIRI observations and is most probably caused by charge trapping (see, for example, ref. 10). We decided to remove the first ten integrations from each visit, corresponding to approximately 6 min or 3% of the data per visit, so that we do not have to also model this initial ramp. Finally, we checked for significant outliers in the final light curves by performing an iterative $5\sigma$ outlier clipping procedure. However, no integrations were removed during this process, leaving us with 288, 287, 287 and 288 integrations for the four visits, respectively.

**Data reduction ED.** For the second data reduction, we also used the Eureka! pipeline[9] for stages 1 to 5. We also started from the uncal.fits files and used the default jwst pipeline settings with the exception of the ramp-fitting weighting parameters in stage 1 that we set to uniform instead of default, as it slightly improved the rms of our residuals. This improvement can be explained by the fact that the default ramp-fitting algorithm uses a weighting of the ramp that gives further weight to the first and last groups of the ramp, which can be problematic when the number of groups is small, such as for TRAPPIST-1 c (only 13 groups). Indeed, the first and last groups can be affected by effects such as the reset switch charge decay or saturation. Thus, to ensure that we fit the ramp correctly, we used an unweighted algorithm that applies the same weight to all groups. Furthermore, in stage 2, we turned off the photom step. Then, in stage 3, we defined a subarray region ([632, 752], [450, 570]), masked the pixels flagged in the DQ array, interpolated bad pixels and performed aperture photometry on the star with an aperture size that minimized the rms of the residuals for each visits. For each integration, we recorded the centre and width of the PSF in the $x$ and $y$ directions after fitting a 2D Gaussian. We computed the background on an annulus of 20–35 pixels (centred on the target) and subtracted it. We note that the choice of the background annulus has little impact on the light curve. We did not remove any integrations a priori but, in stage 4, we sigma clipped $4\sigma$ outliers compared with the median flux calculated using a 10-integrations-width boxcar filter. Then, for each visit for aperture photometry, we chose the aperture radius that led to the smaller rms. These radii were 3.7, 4.0, 3.6 and 3.8 pixels, respectively (see Extended Data Table 2).

**Data reduction MG.** We reduced the data using the following methodology. Starting from the uncal.fits files, we calibrated them using the two first stages of the Eureka! pipeline[9]. We performed a systematic exploration of all the combinations of all Eureka! stage 1 options and we selected the combination resulting in the most precise light curves. Our selected combination corresponds to the default jwst pipeline settings, except for (1) the ramp-fitting weighting parameter set to uniform and (2) the deactivation of the jump correction. The rest of the reduction was done using a pipeline coded in IRAF and Fortran 2003. It included for each calibrated image (1) a change of unit from MJy $sr^{-1}$ to recorded electrons, (2) the fit of a 2D Gaussian function on the profile of the star to measure the subpixel position of its centroid and its full width at half maximum (FWHM) in both directions and (3) the measurement of the stellar and background fluxes using circular and annular apertures, respectively, with IRAF/DAOPHOT[33]. Finally, the resulting light curves were normalized and outliers were discarded from them using a $5\sigma$ clipping with 20-min moving median algorithm. For each visit, the radius of the circular aperture used to measure the stellar flux was optimized by minimizing the standard deviation of the residuals. For each stellar flux measurement, the corresponding error was computed taking into account the star and background photon noise, the readout noise and the dark noise, and assuming a value of 3.1 el $ADU^{-1}$ for the gain (E. Ducrot, private communication). See Extended Data Table 2 for more details.

**Data reduction PT.** We performed an extra analysis using the level 2 (flux-calibrated) 'calints' science products as processed by the Space Telescope Science Institute and hosted on the MAST archive. We determined centroid positions and average seeing FWHM values in the $x$ and $y$ dimensions with a 2D Gaussian fit to the star. We performed fixed-aperture photometry with circular apertures centred on the source centroids, with radii ranging from 3.2 pixels in 0.1-pixel increments. We also performed variable-aperture photometry using circular apertures with radii set to $c$ times a smoothed time series of the measured FWHM values, in which $c$ ranged from 0.75 in increments of 0.05. We smoothed the FWHM values using a 1D Gaussian

kernel with a standard deviation of 2. For both fixed-aperture and variable-aperture photometry, we measured the background using a circular annulus with an inner radius of 30 pixels and an outer radius of 45 pixels. We subtracted the sigma-clipped mean of the pixel values within this annulus from the source counts in each frame, using a clipping level of $4\sigma$. Finally, we recorded the values of a grid of background-subtracted pixels interior to the average photometric aperture size surrounding the source centroid in each frame. We used normalized time series of these pixel values to test whether pixel-level decorrelation (PLD) methods developed for minimizing intrapixel effects in Spitzer Space Telescope data[34] are warranted in the analysis of JWST/MIRI time-series data.

We excluded the first integration of each visit from our analysis as the measured source flux in this exposure was found to be substantially lower than the remainder of the time series for each of the four visits. We checked for outliers in each visit by performing sigma clipping with a threshold of $4\sigma$, but no exposures were flagged with this step. We then selected the aperture size and method (fixed or variable) that minimized the out-of-eclipse scatter for each visit for use in our analysis. We found that fixed-aperture photometry provided the best performance in each case, with optimal radii of 4.4, 4.1, 3.9 and 3.5 pixels for the four visits, respectively.

## Data analysis

We fitted each of the reductions to extract an eclipse depth value. The different assumptions for the four global fits are listed in Extended Data Table 3.

**Data analysis SZ.** We fitted the eclipse light curve using the open-source Python MCMC sampling routine emcee[35]. Our full fitting model, $F(t)$, was the product of a batman[36] eclipse model, $F_{eclipse}(t)$, and a systematic model, $F_{sys}(t)$. We fit the systematics of JWST with a model of the following form:

$$F_{sys}(t) = F_{polynom}(t)F_x(t)F_y(t)F_{\sigma_x}(t)F_{\sigma_y}(t),$$

in which $F_{polynom}$ is a polynomial in time and $F_x(t)$, $F_y(t)$, $F_{\sigma_x}(t)$ and $F_{\sigma_y}(t)$ detrend the light curve against a time series of the centroid in $x$ and $y$ and the width of the PSF in $x$ and $y$, respectively. Before fitting the full light curve consistent out of the four visits, we first determined the best systematic model for each visit by minimizing the Bayesian information criterion (BIC)[37–39]. We tried a range of polynomials ranging from zeroth order to third order and detrended for the shift in $x$-pixel and $y$-pixel positions or for the change in the width of the PSF in time. The best final combination of polynomials and detrending parameters for each visit are listed in Extended Data Table 3. Our eclipse model used the predicted transit times from ref. 40, which accounts for the transit-timing variations (TTVs) in the system and we allowed for a non-zero eccentricity. We also accounted for the light travel time, which is approximately 16 s for TRAPPIST-1 c, that is, its semi-major axis is about 8 light-seconds. We fixed the other parameters of the planet and system, such as the ratio of the semi-major axis to stellar radius $a/R_*$, the ratio of the planetary radius to stellar radius $R_p/R_*$ and the inclination $i$, to the values reported in ref. 40. We decided to also supersample the light curve by a factor of 5 in our fitting routine because the sampling of the data ($\approx$every 40 s) is comparable with the ingress/egress duration of 200 s (ref. 40). Our global fit consisted of 32 free parameters: six physical (the eccentricity, the argument of periastron and an eclipse depth for each visit), 22 parameters to fit for the systematics and four free parameters that inflated the uncertainties in the flux for each visit. The four free parameters are necessary because the current gain value on the Calibration References Data System (CRDS) has been empirically shown to be wrong for MIRI data[10]. For our global MCMC, we used 128 walkers (=4 times the number of free parameters), 150,000 steps and discarded the first 20% of steps (=30,000 steps) as burn-in. This

corresponds to approximately 80 times the autocorrelation length. After calculating a weighted average of the four eclipse depths, we get an eclipse depth of $f_p/f_* = 431^{+97}_{-96}$ ppm for this reduction. Extended Data Fig. 4 shows the Allan deviation plots of the residuals for each of the visits and the global fit. The rms of the residuals as a function of bin size follows the inverse square root law, which is expected for Gaussian noise.

**Data analysis ED.** Once we obtained the light curve for each visit from stage 4 of the Eureka! pipeline, we used the Fortran code trafit, which is a revised version of the adaptive MCMC code described in refs. 41–43. It uses the eclipse model in ref. 44 as a photometric time series, multiplied by a baseline model to represent the other astrophysical and instrumental systematics that could produce photometric variations. First, we fit all visits individually. We tested a large range of baseline models to account for different types of external sources of flux variation/modulation (instrumental and stellar effects). This includes polynomials of variable orders in time, background, PSF position on the detector $(x, y)$ and PSF width (in $x$ and $y$). Once the baseline was chosen, we ran a preliminary analysis with one Markov chain of 50,000 steps to evaluate the need for rescaling the photometric errors through the consideration of a potential underestimation or overestimation of the white noise of each measurement and the presence of time-correlated (red) noise in the light curve. After rescaling the photometric errors, we ran two Markov chains of 100,000 steps each to sample the probability density functions of the parameters of the model and the physical parameters of the system and assessed the convergence of the MCMC analysis with the Gelman and Rubin statistical test[45]. For each individual analysis, we used the following jump parameters with normal distributions: $M_*$, $R_*$, $T_{eff,*}$, [Fe/H], $t_0$, $b$; all priors were taken from ref. 46 except for the transit timings, which were derived from the dynamical model predictions by ref. 40. We fixed $P$, $i$ and $e$ to the literature values given in refs. 40,46. The eclipse depths that we computed for each visit individually were $445 \pm 193$ ppm, $418 \pm 173$ ppm, $474 \pm 158$ ppm and $459 \pm 185$ ppm, respectively.

We then performed a global analysis with all four visits, using the baseline models derived from our individual fits for each light curve. Again, we performed a preliminary run of one chain of 50,000 steps to estimate the correction factors that we then apply to the photometric error bars and then a second run with two chains of 100,000 steps. The jump parameters were the same as for the individual fits except for the fact that we fixed $t_0$ and allowed for TTVs to happen for each visit (each transit TTV has an unconstrained uniform prior). We used the Gelman and Rubin statistic to assess the convergence of the fit. We measure an eclipse depth of $423^{+97}_{-95}$ ppm from this joint fit.

**Data analysis MG.** Our data-analysis methodology was the same as that used by ED, that is, we used the Fortran 2003 code trafit to perform a global analysis of the four light curves, adopting the Metropolis–Hasting MCMC algorithm to sample posterior probability distributions of the system's parameters. Here too, we tested for each light curve a large range of baseline models and we adopted the ones minimizing the BIC. They were (1) a linear polynomial of time for the first visit, (2) a cubic polynomial of time and a linear polynomial of the $y$ position for the second visit, (3) a linear polynomial of time and of the $x$ position for the third visit and (4) a cubic polynomial of time and of the $y$ position for the fourth visit. We also performed a preliminary analysis (composed of one Markov Chain of 10,000 steps) to assess the need to rescale the photometric errors for white and red noise. We then performed two chains of 500,000 steps each (with the first 20% as burn-in). The convergence of the analysis was checked using the Gelman and Rubin statistical test[45]. The jump parameters of the analysis, that is, the parameters perturbed at each step of the MCMC chains, were (1) for the star, the logarithm of the mass, the logarithm of the density, the effective temperature

and the metallicity and (2) for the planet, the planet-to-star radius ratio, the occultation depth, the cosinus of the orbital inclination, the orbital parameters $\sqrt{e}\cos\omega$ and $\sqrt{e}\sin\omega$ (with $e$ the orbital eccentricity and $\omega$ the argument of pericentre) and the timings of the transits adjacent to each visit. We assumed normal prior distributions for the following parameters based on the results from ref. 40: $M_* = 0.0898 \pm 0.023$, $R_* = 0.1192 \pm 0.0013$, $T_{eff} = 2{,}566 \pm 26$ K and [Fe/H] = 0.05 ± 0.09 for the star; $(R_p/R_*)^2 = 7{,}123 \pm 65$ ppm, $b = 0.11 \pm 0.06$ and $e = 0 + 0.003$ (semi-Gaussian distribution) for the planet. We also tested the assumption of a circular orbit and obtained similar results. For each visit, we considered for the timings of the two adjacent transits normal prior distributions based on the predictions of the dynamical model of ref. 40. At each step of the MCMC, the orbital position of the planet could then be computed for each time of observation from the timings of the two adjacent transits and from $e$ and $\omega$ and taking into account the approximately 16 s of light-travel time between occultation and transit. This analysis led to the value of 414 ± 91 ppm for the occultation and to an orbital eccentricity of $0.0016^{+0.0015}_{-0.0008}$, consistent with a circular orbit. Under the assumption of a circular orbit, our analysis led to an occultation depth of 397 ± 92 ppm, in excellent agreement with the result of the analysis assuming an eccentric orbit.

We also performed a similar global analysis but allowing for different occultation depths for each visit. The resulting depths were 400 ± 163 ppm, 374 ± 184 ppm, 421 ± 187 ppm and 403 ± 202 ppm, that is, they were consistent with a stable thermal emission of the planet's dayside (at this level of precision). Similar to data reduction SZ, we also did create Allan deviation plots for this particular data reduction. The best-fit residuals as a function of bin size from each visit do generally follow the inverse square root law (see Extended Data Fig. 4 for the Allan deviation plots of data reduction SZ).

Finally, we computed the brightness temperature of the planet at 15 μm from our measured occultation depth using the following methodology. We measured the absolute flux density of the star in all the calibrated images, using an aperture of 25 pixels, large enough to encompass the wings of its PSF. We converted these flux densities from MJy sr⁻¹ to mJy and computed the mean value of 2.559 mJy and the standard deviation of 0.016 mJy. We added quadratically to this error of around 0.6% a systematic error of 3%, which corresponds to the estimated absolute photometric precision of MIRI (P.-O. Lagage, private communication). It resulted in a total error of 0.079 mJy. Multiplying the measured flux density by our measured occultation depth led to a planetary flux density of 1.06 ± 0.23 μJy. Multiplying again this result by the square of the ratio of the distance of the system and the radius of the planet and dividing by π led to the mean surface brightness of the planet's dayside. Applying Planck's law, we then computed the brightness temperature of the planet, whereas its error was obtained from a classical error propagation. Our result, for this specific reduction, was 379 ± 30 K, to be compared with an equilibrium temperature of 433 K computed for a null-albedo planet with no heat distribution to the nightside.

It is also worth mentioning that applying the same computation on the star itself led to a brightness temperature of 1,867 ± 55 K, which is much lower than its effective temperature.

**Data analysis PT.** We began our analysis by determining which time-series regressors (if any) should be included for fitting systematics in the photometry on the basis of the BIC. Our total model is the product of a batman eclipse model ($F_{eclipse}$) and a systematics model ($F_{syst}$) to the data, which has a general form of

$$F_{syst}(t) = F_{polynom}(t) F_x(t) F_y(t) F_{FWHM}(t) F_{ramp}(t) F_{PLD}(n, t).$$

Here $F_{polynom}$ is a polynomial in time, $F_x(t)$ and $F_y(t)$ are time series of the target centroids in $x$ and $y$, respectively, $F_{FWHM}(t)$ is the time series

of average FWHM values for the source determined with a 2D Gaussian fit and $F_{ramp}$ is an exponential function that accounts for ramp-up effects. $F_{PLD}(n, t)$ is the linear combination of $n$ basis pixel time series and it has a form of

$$F_{PLD}(n, t) = \sum_{i=1}^{n} C_i \hat{P}_i(t)$$

Here $\hat{P}_i(t)$ is the normalized intensity (from 0) of pixel $i$ at time $t$ and $C_i$ is the coefficient of pixel $i$ determined in the fit. PLD was developed to mitigate systematic intrapixel effects in Spitzer/Infrared Array Camera (IRAC) data[34], in which the combination of source PSF motion and intrapixel gain variations introduced percent-level correlated noise in time-series data (for example, ref. 47).

In our analysis, we tested forms of $F_{polynom}$ ranging from degree 0 and different sets of PLD basis pixels, including the brightest 1, 4, 9, 16, 25 and 36 pixels. For each visit, we explored grids of every possible combination of the components of $F_{syst}(t)$. For each combination, we first initialized the coefficients of each component using linear regression. We then used emcee to perform an MCMC fit of the total eclipse and systematic model to the visit data. We ran $2v + 1$ walkers for 10,000 steps in each fit, in which $v$ represents the number of free parameters in the total model. The first 1,000 steps of these chains were discarded as burn-in. We fit for seven physical parameters in our calculation of $F_{eclipse}$, these being the orbital period, $a/R_*$, orbital inclination, eccentricity, longitude of periastron, eclipse depth and time of secondary eclipse. Gaussian priors were assigned to these parameters with means and standard deviations set by their measurements reported in ref. 40. We also placed Gaussian priors on the coefficients of the components of $F_{syst}$, with means set by the linear regression fit and standard deviations set to the absolute value of the square root of those values.

We calculated the BIC of the best-fitting model that resulted from the MCMC analysis and then selected the form of $F_{syst}$ that minimized the BIC. The form of $F_{syst}$ that we determined for each visit with this approach consisted of only an $F_{polynom}$ component. The first visit was best fit by a linear polynomial, whereas the remaining three were best fit by a quadratic polynomial.

With the form of $F_{syst}(t)$ determined for each visit, we then performed a joint fit of all four eclipses. This fit included 18 total free parameters: seven physical and 11 for fitting systematics (see Extended Data Table 3). We ran this fit with 64 chains for 50,000 steps, discarding the first 5,000 steps for burn-in. We measured a resulting eclipse depth of $418^{+90}_{-91}$ ppm from this fit.

**Brightness temperature calculation**
The following analysis was based on stage 0 (.uncal) data products pre-processed by the JWST data processing software version number 2022_3b and calibrated with Eureka! as described above in the 'Data reduction MG' section. We computed the brightness temperature of the planet at 15 μm from our measured occultation depth using the following methodology. We measured the absolute flux density of the star in all the calibrated images, using an aperture of 25 pixels, large enough to encompass the wings of its PSF. We converted these flux densities from MJy sr⁻¹ to mJy and computed the mean value of 2.559 mJy and the standard deviation of 0.016 mJy. We added quadratically to this error of around 0.6% a systematic error of 3%, which corresponds to the estimated absolute photometric precision of MIRI (P.-O. Lagage, private communication). It resulted in a total error of 0.079 mJy. Multiplying the measured flux density by our measured occultation depth led to a planetary flux density of 1.06 ± 0.23 μJy. Multiplying again this result by the square of the ratio of the distance of the system and the radius of the planet and dividing by π led to the mean surface brightness of the planet's dayside. Applying Planck's law, we then computed the brightness temperature of the planet, whereas its error was obtained from a classical error propagation. Our result, for the MG reduction,

was 379 ± 30 K, to be compared with an equilibrium temperature of 433 K computed for a null-albedo planet with no heat distribution to the nightside. It is also worth mentioning that applying the same computation on the star itself led to a brightness temperature of 1,867 ± 55 K, which is significantly lower than its effective temperature.

### Emission modelling for TRAPPIST-1 c
We generated various emission spectra for TRAPPIST-1 c to compare them to our measured eclipse depth at 15 μm. These models include (1) bare-rock spectra, (2) $O_2/CO_2$ mixture atmospheres and pure $CO_2$ atmospheres and (3) coupled climate–photochemical forward models motivated by the composition of Venus. In the following, we describe each of these models.

**Bare rock.** Our bare-rock model is a spatially resolved radiative transfer model and computes scattering and thermal emission for a variety of surface compositions. For each composition, the radiative equilibrium temperature of the surface is computed on a 45 × 90 latitude–longitude grid, assuming that TRAPPIST-1 c is tidally locked. Surface reflectance and emissivity data are from ref. 22, which were derived from reflectance spectra of rock powders or minerals measured in the laboratory combined with an analytical radiative-transfer model[48]. These data have previously been used to model surface albedos and emission spectra of bare-rock exoplanets[7,22,49]. Here we consider six compositions as well as a blackbody: basaltic, feldspathic, Fe-oxidized (50% nanophase haematite, 50% basalt), granitoid, metal-rich ($FeS_2$) and ultramafic (see Extended Data Fig. 5). Given the uncertainty in the measured eclipse depth, we assume a Lambertian surface with isotropic scattering and emission and neglect the angular dependency of the surface reflectance and emissivity, which would depend on the surface roughness and regolith particle size[22]. Sensitivity tests show that these surface-model assumptions are indistinguishable within the current precision of the TRAPPIST-1 c measurements (not shown).

Furthermore, albedos and spectra of bare rocks in the Solar System are modified by space weathering, so we also consider the impact of space weathering on TRAPPIST-1 c. The timescale for lunar space weathering through exposure to the solar wind has been estimated to range from about $10^5$ years to about $10^7$ years (refs. 50,51). We extrapolate from the lunar value to TRAPPIST-1 c using scaling relations from a stellar-wind model[52]. We find that the space-weathering timescale for TRAPPIST-1 c is much shorter than the lunar value, about $10^2$–$10^3$ years, largely because of the planet's small semi-major axis. An exposed surface on TRAPPIST-1 c should therefore have been substantially weathered. To simulate the impact of space weathering on unweathered surfaces, we incorporate the same approach as that in refs. 23,53. The surface composition is modelled as a mixture of a fresh host material (described above) and nanophase metallic iron using Maxwell–Garnett effective medium theory. The refractive index of metallic iron is taken from ref. 54.

**Simple 1D $O_2/CO_2$ mixtures.** We construct a grid of $O_2$-dominated model atmospheres with a range of surface pressures and mixing ratios of $CO_2$. These are broadly representative of a plausible outcome of planetary atmosphere evolution, in which water in the atmospheres of terrestrial planets orbiting late-type M dwarfs is photolysed and the H is lost, leaving a large $O_2$ reservoir[15,55]. The atmosphere models we construct are 1D models following the approach presented in ref. 2, with adiabatic pressure–temperature profiles in the deep atmosphere and isothermal pressure–temperature profiles above 0.1 bar (for thicker atmospheres, $P > 0.1$ bar) or the skin temperature (for thinner atmospheres). This approach uses DISORT[56,57] to calculate radiative transfer in 1D through the atmosphere to generate emission spectra.

We do consider how an atmosphere can transport heat to the nightside. To include heat transport to the nightside, we implement the analytic approach in ref. 14; we use the redistribution factor $f$ calculated in equation (3) of that work for each of the models in the grid. We assume

that both the surface Bond albedo and the top-of-the-atmosphere Bond albedo are 0.1. We construct a grid of $O_2$-dominated model atmospheres with surface pressures from 0.01 to 100 bar (in 1-dex steps) and $CO_2$ mixing ratios from 1 ppm to 10,000 ppm (in 1-dex steps). We also generate pure $CO_2$ atmospheres with the same surface pressures. For the thicker atmospheres ($P_{surf} \geq 1$ bar), we set the thermopause (in which the atmosphere transitions from adiabatic to isothermal) to 0.01 bar.

**Coupled climate–photochemical Venus-like atmospheres.** We use a 1.5D coupled climate–photochemical forward model, VPL Climate[3,58,59], that explicitly models day and night hemispheres with layer-by-layer, day–night advective heat transport driven by simplified versions of the 3D primitive equations for atmospheric transport to simulate plausible atmospheric states for TRAPPIST-1 c for cloudy Venus-like scenarios. VPL Climate uses SMART[60] with DISORT[56,57] for spectrum-resolving radiative transfer for accuracy and versatility for both the climate modelling and the generation of the resulting planetary spectra. The model has been validated for Earth[61] and Venus[62] but is capable of modelling a range of atmospheric states.

Owing to the early luminosity evolution of the star, TRAPPIST-1 c would have been subjected to very high levels of radiation[63] and so we would anticipate evolved atmospheres that had undergone atmospheric and possibly ocean loss[6]. We start with the self-consistently coupled climate–photochemical Venus-like atmospheres generated for an evolved TRAPPIST-1 c from ref. 3, with 96.5% $CO_2$ and 3.5% $N_2$ and assume Venus lower atmospheric trace gases and self-consistent generated sulfuric acid aerosols. We use these atmospheres as a starting point for 1.5D clear-sky Venus-like atmospheres (0.1, 1 and 10 bar) and 1.5D cloudy Venus-like atmospheres (10 bar) with sulfuric acid haze. Note that 10-bar Venus-like atmospheres will produce similar results to a 93-bar Venus-like atmosphere owing to the emitting layer being above or at the cloud deck, which is at a similar pressure for the 10-bar and 93-bar cases. All the modelled clear-sky Venus atmospheres produce 15-μm $CO_2$ features with depths spanning 134–143 ppm, with the cloudy 10-bar Venus centred at 181 ppm. Because $H_2SO_4$ aerosols are likely to condense in the atmosphere of a Venus-like planet at the orbital distance of TRAPPIST-1 c (ref. 3), we show the dayside spectrum for the 10-bar cloudy Venus for comparison with the data in Fig. 3. The emitting layer (cumulative optical depth 1) for the cloud aerosols occurs at 7 mbar in this atmosphere, although the 15-μm $CO_2$ absorption is sufficiently strong that it emits from a comparable pressure level in the core of the band. The observations rule out a self-consistent Venus-like atmosphere for TRAPPIST-1 c to 2.6$\sigma$.

### Atmospheric escape models
We use energy-limited atmospheric escape models[6,15] from a steam atmosphere to explore the amount of atmospheric escape that TRAPPIST-1 c may have experienced over its lifetime. The model assumes that escape occurs in the stoichiometric ratios of H/O in water vapour, allows for escape of oxygen and reaction of oxygen with the magma ocean. The model transitions from magma ocean to passive stagnant-lid outgassing when surface temperatures drop below the silicate melting point. Escape continues throughout all tectonic stages. In Fig. 4, we show the final amount of $O_2$ gas left in the atmosphere after 7.5 Gyr of evolution for a range of planetary water abundances and XUV saturation fractions. For typical saturation fractions of $10^{-3}$ (refs. 25,64), our observations suggest that the planet probably had a relatively low starting volatile abundance. We note that these models are probably upper limits on thermal escape and more detailed models of escape, especially incorporating other gases such as $CO_2$ and $N_2$, are needed in the future to confirm these results. We also estimate total ion-driven escape fluxes resulting from stellar-wind interactions of a minimum of 1–3 bar over the lifetime of the planet, assuming constant stellar wind over time[65]. We also considered the extended pre-main sequence for a star such as TRAPPIST-1. We used the stellar evolution models of ref. 63 for a $0.09 M_\odot$ star to approximate the pre-main sequence evolution of the star.

## Interior-structure model

We use an interior-structure model to perform an MCMC retrieval on the planetary mass and radius of TRAPPIST-1 c and the possible stellar Fe/Si of TRAPPIST-1. The estimated Fe/Si mole ratio of TRAPPIST-1 is $0.76 \pm 0.12$ (ref. 66), which is lower than the solar value, Fe/Si = 0.97 (ref. 67). Our interior-structure model solves a set of differential equations to compute the density, pressure, temperature and gravity as a function of radius in a one-dimensional grid[68,69]. The interior model presents two distinct layers: a silicate-rich mantle and an Fe-rich core. On top of the mantle, we couple the interior model with an atmospheric model to compute the emission and the Bond albedo. These two quantities enable us to solve for radiative-convective equilibrium, find the corresponding surface temperature at the bottom of the atmosphere and find the total atmospheric thickness from the surface up to a transit pressure of 20 mbar (refs. 20,70). We consider an $H_2O$-dominated atmosphere, with 99% $H_2O$ and 1% $CO_2$. Our 1D, $k$-correlated atmospheric model prescribes a pressure–temperature profile comprising a near-surface convective layer and an isothermal region on top. In the regions of the atmosphere in which the temperature is low enough for water to condense and form clouds, we compute the contribution of these to the optical depth and their reflection properties as described in refs. 71,72.

The posterior distribution function of the surface pressure retrieved by our MCMC indicates a $1\sigma$ confidence interval of $40 \pm 40$ bar for TRAPPIST-1 c. Surface pressures between 0 and 120 bar would be compatible with our probability density function within $2\sigma$ (ref. 73). Oxygen is more dense than $H_2O$. Consequently, for a similar surface pressure, an $O_2$-rich atmosphere would be less extended than the $H_2O$-dominated envelope we consider in our coupled interior-atmosphere model. This means that the density of TRAPPIST-1 c could be reproduced with an $O_2$-rich atmosphere with a surface pressure as low as our $H_2O$ upper limit, 80 bar.

## Stellar properties

The stellar properties of TRAPPIST-1 have been constrained with observations of the total luminosity of the star, $L_* = 4\pi D_*^2 F_{bol}$ (based on broadband photometry to obtain the bolometric flux of the star, $F_{bol}$, and a distance measured with GAIA, $D_*$), a mass–luminosity relation[74] to obtain the stellar mass, $M_*(L_*)$, with uncertainty, as well as a precise stellar density, $\rho_*$, thanks to modelling of the seven transiting planets[40,46,75]. These combine to give the stellar radius $R_*$ and effective temperature $T_{eff,*}$,

$$R_* = \left(\frac{3M_*}{4\pi\rho_*}\right)^{1/3} \propto M_*^{1/3},$$

$$T_{eff} = \left(\frac{L_*}{4\pi R_*^2 \sigma}\right)^{1/4} \propto M_*^{-1/6}.$$

The properties of the planets have also been measured precisely in relation to the star using the depths of transit, yielding $R_p/R_*$, and transit-timing variations, yielding $M_p/M_*$ (ref. 40). To convert the secondary eclipse depth, $\delta = F_p/F_*$, into a brightness temperature of the planet requires an estimate of the brightness temperature of the star:

$$\delta = \frac{I_{b,p}}{I_{b,*}}\frac{R_p^2}{R_*^2},$$

or

$$I_{b,p} = I_{b,*}\delta\frac{R_*^2}{R_p^2},$$

in which $I_{b,*}$ and $I_{b,p}$ are the mean surface brightness of the star and planet in the MIRI band at full phase (that is, secondary eclipse), respectively. The ratio $R_p/R_*$ is well constrained from the transit depth, whereas the brightness temperature of the star can be measured with an absolute calibration of the stellar flux in the MIRI band, $F_*$ (for example, ref. 46). The stellar intensity may then be computed as:

$$I_{b,*} = \frac{F_*}{\Omega_*} = \frac{F_* D_*^2}{\pi R_*^2},$$

in which $\Omega_*$ is the solid angle of the star. Because our estimate of $R_*$ is proportional to $M_*^{1/3}$, this means that $I_{b,*} \propto M_*^{-2/3}$. For a given value of $R_*$, this surface brightness can be translated into a brightness temperature, $T_{b,*}$, and with the equation above, we can compute the intensity and therefore the surface brightness of the planet, $T_{b,p}$, to be $380 \pm 31$ K using the eclipse depth and the stellar flux density. We also estimate the stellar brightness temperature in the MIRI band with an atmospheric model for the star relating $T_{b,*}$ in the MIRI band to the $T_{eff}$, as $\alpha = T_{b,*}/T_{eff,*}$. We have accomplished this with the state-of-the-art SPHINX model for low-temperature stars[76] and assumed $T_{eff} = 2,566$ K (ref. 40), yielding $\alpha = 0.72$ at 14.87 μm. We also compute the $\alpha$ from JWST spectrophotometric observations with a flux of $2.599 \pm 0.079$ mJy at 14.87 μm, yielding $\alpha = 0.71 \pm 0.02$. The MIRI images are flux-calibrated (with an internal error of 3%). We measure the stellar flux in all images within an aperture large enough to encompass the whole PSF and then compute the mean and the standard deviation. We compute the total error on the measurement to be 3% larger than this standard deviation. As the unit of flux in MIRI images is given in Jy str$^{-1}$, we multiply the measured fluxes by the angular area covered by a pixel in str to yield units of Jy.

The stellar brightness temperature scales linearly with effective temperature and metallicity in the MIRI wavelength range and scales inversely with surface gravity of the star. The effective temperature, however, scales as $T_{eff} \propto M_*^{-1/6}$ (or $R_*^{-1/2}$), with stellar mass (or radius) relative to the estimate based on the measured flux. The estimate of $\alpha$, therefore, may have a substantial imprecision given the possible heterogeneity of the stellar atmosphere, as well as the inherent uncertainties involved in modelling late-type stellar atmospheres. Both the synthetically derived $\alpha$ and those from observations match within $2\sigma$ uncertainty, lending credence to empirical mass–luminosity relations and synthetic atmosphere-model-derived stellar brightness temperatures. Note, however, that the mass–luminosity relation is only calibrated with a handful of low-mass stars in binaries[74] and hence its applicability to TRAPPIST-1 may be tenuous; this may thus be the weakest link in determining the stellar parameters. Assumption-driven deviations between synthetic models for late-type stars and empirically calibrated methods both still remain a notable challenge in truly understanding these hosts.

## Eclipse-timing variations

Dynamical modelling of the TRAPPIST-1 system[40] gives a precise forecast of the times of transit and eclipse for all seven planets. These have been used in the planning of the observations and can also be compared with the measured times.

The times of eclipse can be offset from the mid-point between the times of transit owing to four different effects: (1) the light-travel time across the system[77], (2) non-zero eccentricity[78], (3) non-uniform emission from an exoplanet[79] (this does not change the mid-point of the eclipse but it does change the shape of ingress/egress and can lead to an artificial time offset if not accounted for in the modelling) and (4) eclipse-timing variations owing to perturbations by other planets in the system. Of these three effects, the second effect is typically the largest, which can be used to constrain one component of the eccentricity vector of the transiting planet[78].

In Extended Data Table 4, we list the measured eclipse times from the four different reductions and in Extended Data Fig. 6 we compare them with the forecast from ref. 40. To make the forecast, we used the posterior probability of the timing model to compute the times of transit and eclipse and then we calculate the time of eclipse minus the mean of the two adjacent transits of planet c to derive an 'eclipse-timing offset'. This offset should be zero for a circular, unperturbed orbit with negligible light-travel time (which is about 16 s, or $1.8 \times 10^{-4}$ days for TRAPPIST-1 c). The dynamical modelling is constrained by the times of transit, which place some constraint on the eccentricity of the orbit of planet c (in particular, the mean or free eccentricity could be non-zero). The uncertainty on the eccentricity leads to uncertainty on the times of secondary eclipse. Our forecast models for the eclipse-timing offset have a $1\sigma$ uncertainty of about 3.5 min at the measured times of eclipse (approximately 0.0024 days).

The measured times were taken from four analyses (by SZ, PT, ED and MG), in which a broad prior was placed on the times of transit, whereas the duration and depth were constrained to the measured values of the four eclipses. The times of each eclipse were then free to vary and the posterior times of transit were inferred using MCMC (ED/MG/PT) or nested sampling (SZ). The four analyses give good agreement on the values but have substantial differences between the uncertainties.

Overall, the forecast eclipse-timing offsets agree well with the measured times, within 1–2$\sigma$ offsets. The uncertainties on the measured times are comparable with the forecast uncertainties and so, in future work, we hope to use these measured eclipse times to further constrain the eccentricity vector of the orbit of planet c. This may help to constrain tidal damping models of planet c but it may also constrain tidal damping of all of the planets, as the free eccentricity vector of planet c is tightly correlated with those of the other planets owing to the 'eccentricity–eccentricity' degeneracy present in transiting planet systems[80].

## Data availability

The data used in this work were collected by the James Webb Space Telescope as part of General Observer programme 2304 and will be publicly accessible after the default proprietary period of one year. The most recently taken visit will therefore be publicly accessible on the Mikulski Archive for Space Telescopes on 1 December 2023.

## Code availability

We used the following codes, resources and Python packages to reduce, analyse and interpret our JWST observations of TRAPPIST-1 c: numpy[81], matplotlib[82], astropy[83], batman[36], Eureka![9], jwst[32], emcee[35], trafit[41–43], dynesty[84,85], SMART[60], VPL Climate[3,58,59], DISORT[56,57] and IRAF/DAOPHOT[33]. We can share the code used in the data reduction or data analysis on request.

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

**Acknowledgements** This work is based in part on observations made with the NASA/ESA/CSA James Webb Space Telescope. The data were obtained from the Mikulski Archive for Space Telescopes at the Space Telescope Science Institute, which is operated by the Association of Universities for Research in Astronomy, Inc., under NASA contract NAS 5-03127 for JWST. These observations are associated with programme 2304. M.G. is F.R.S.-FNRS Research Director and acknowledges support from the Belgian Federal Science Policy Office (BELSPO) BRAIN 2.0 (Belgian Research Action through Interdisciplinary Networks) for the project PORTAL no. B2/212/P1/PORTAL (PhOtotrophy on Rocky habiTAble pLanets). V.S.M. and A.P.L. are part of the Virtual Planetary Laboratory Team, which is a member of the NASA Nexus for Exoplanet System Science, and financed through NASA Astrobiology Program grant 80NSSC18K0829. A.R.I. acknowledges support from the NASA FINESST grant 80NSSC21K1846.

**Author contributions** S.Z., L.K., M.G., P.T., E.D., A.P.L., V.S.M., D.D.B.K., C.M., L.S., E.A., L.A. and G.S. contributed notably to the writing of this manuscript. S.Z., E.D., P.T. and M.G. provided a data reduction and data analysis of the four visits for this work and contributed an eclipse depth value. C.M., D.D.B.K., X.L., R.H., A.P.L. and V.S.M. ran theoretical models for the planet's atmosphere and surface. L.A. ran models on the planet's interior structure. A.R.I. and E.A. modelled the stellar spectrum. L.S. modelled the atmospheric escape for the planet. L.K., M.G., V.S.M., D.D.B.K., R.H., C.M., L.S., E.A., F.S., E.B. and A.M.M. contributed to the observing proposal. E.D. is Paris Region Fellow, Marie Sklodowska-Curie Action.

**Funding** Open access funding provided by Max Planck Society.

**Competing interests** The authors declare no competing interests.

**Additional information**
**Correspondence and requests for materials** should be addressed to Sebastian Zieba.

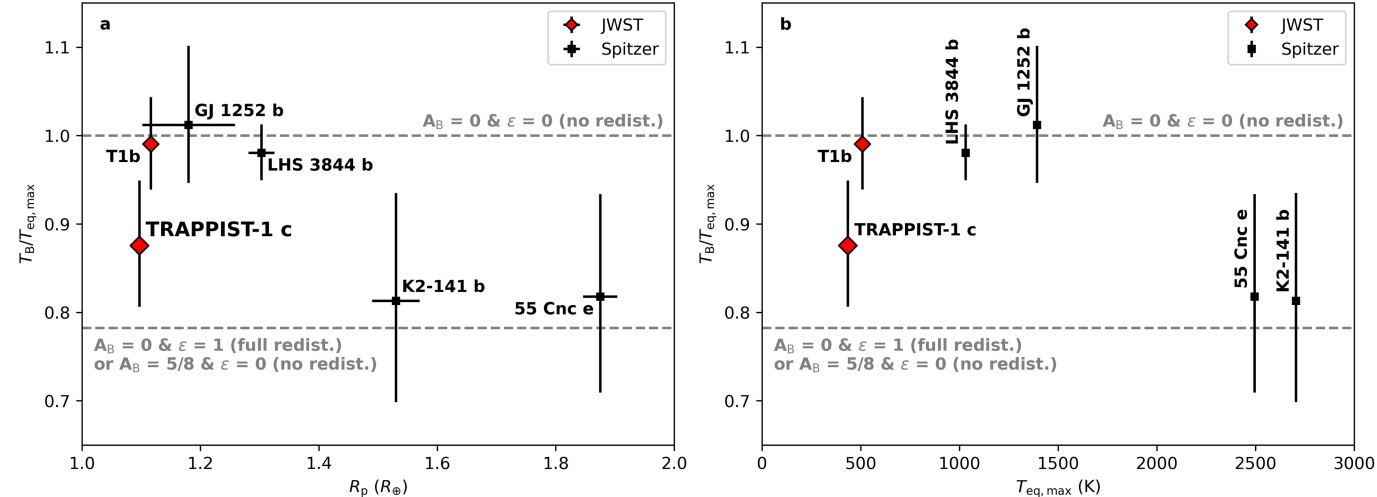

**Extended Data Fig. 1 | Comparison of small exoplanets with measured infrared emission.** Following ref. 8, we show the normalized dayside brightness temperature for super-Earths ($R_p < 2R_\oplus$) with measured thermal emission, as a function of planet size (**a**) and maximum equilibrium temperature (**b**). The brightness temperatures are normalized relative to predictions for a bare rock with zero albedo and zero heat redistribution, $T_{eq,max}$. The thermal emission of TRAPPIST-1 c has been detected in this work at 15 μm. The other planets are TRAPPIST-1 b (T1b in the plot; also at 15 μm) and planets that have been observed with Spitzer's IRAC channel 2 at 4.5 μm. The uncertainties on the radius for the planets in the TRAPPIST-1 system are smaller than the marker symbol. Error bars show 1σ uncertainties.

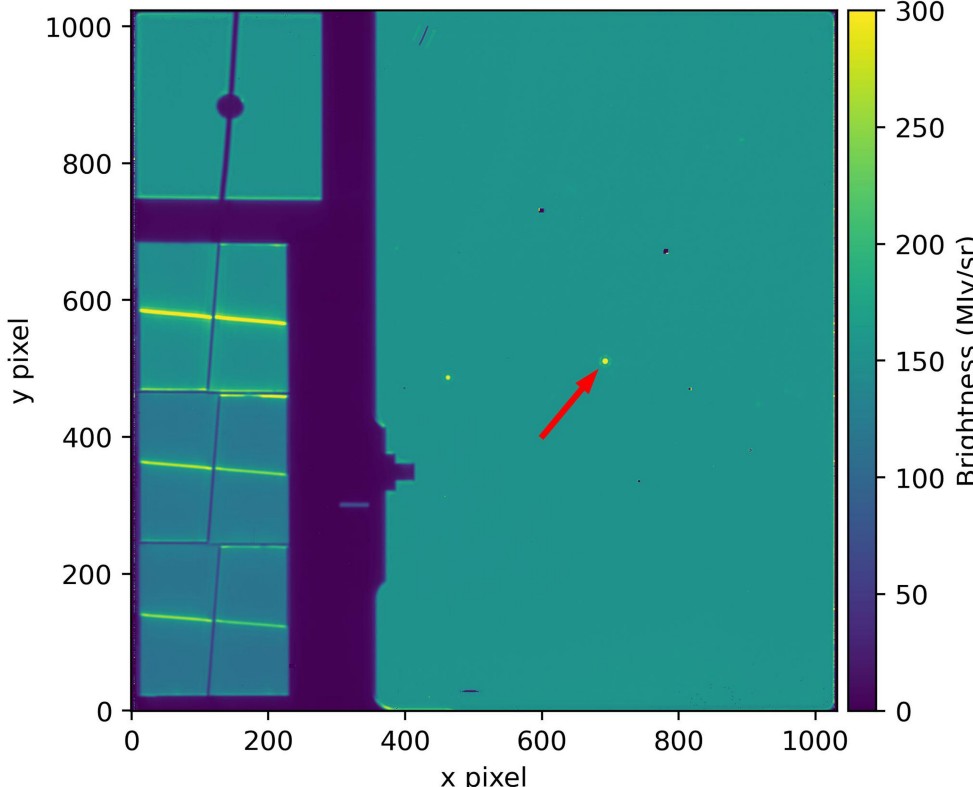

**Extended Data Fig. 2 | Example of a MIRI integration using the FULL array.** An integration taken during our observations showing the MIRI imager focal plane. Most of the FULL array is taken up by the imager field of view on the right side. TRAPPIST-1 is centred on the imager highlighted by the red arrow. The left side of the imager was not used in our analysis and consists of the Lyot coronagraph (top left) and the three four-quadrant phase masks coronagraphs (lower left).

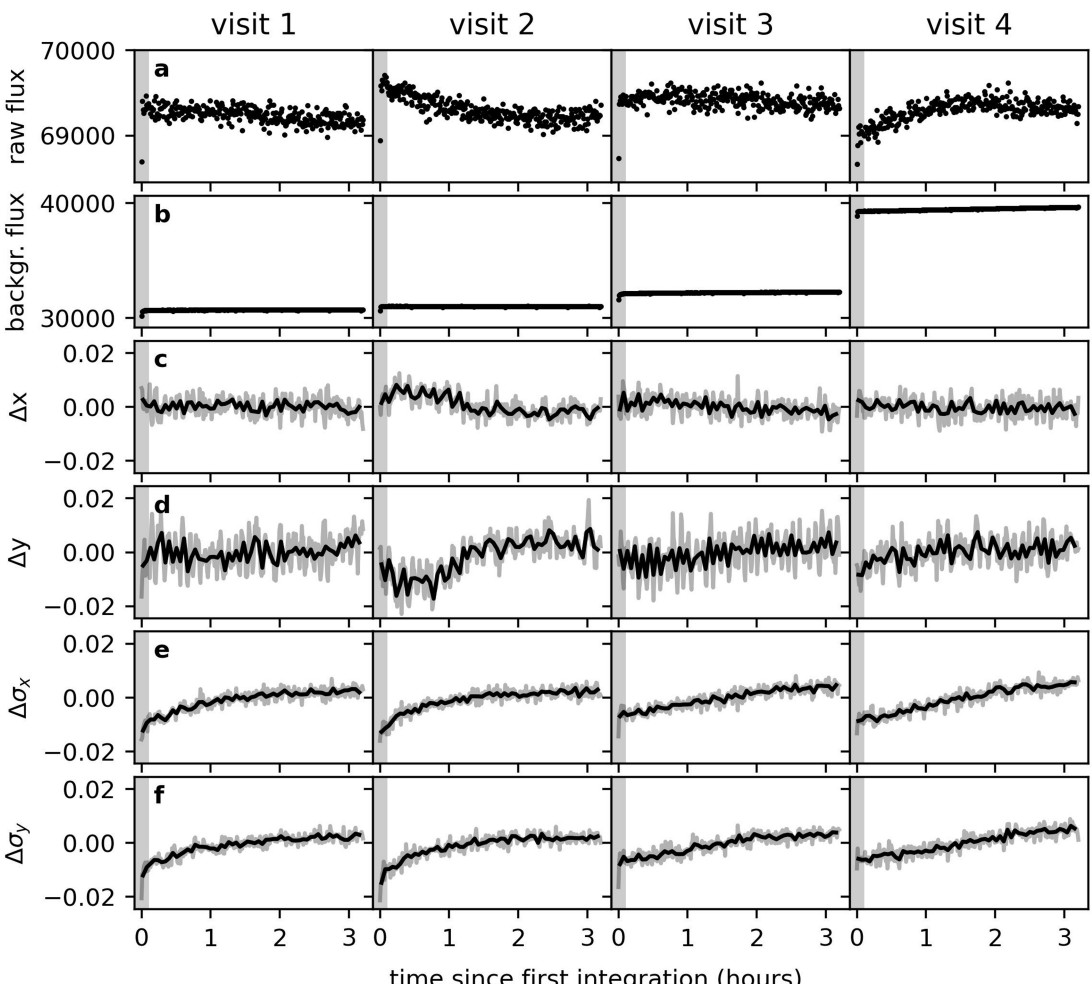

**Extended Data Fig. 3 | Diagnostic plot of all four visits taken during JWST General Observer programme 2304 based on the SZ reduction.** Every column corresponds to a visit. **a**,**b**, The top and second rows show the raw and background flux in units of electrons per integration per pixel, respectively. The raw flux is referring to the flux level within the target aperture before the subtraction of the background flux. **c**–**f**, The following rows are depicting the properties of the centroid over time. We fitted a 2D Gaussian distribution to the target at every integration to determine its $x$ and $y$ positions on the detector. $\Delta\sigma_x$ and $\Delta\sigma_y$ describe change in the width of the 2D Gaussian with time. The integrations were taken approximately every 40 s. The lower four rows were also binned to 5 min (=8 integrations) shown with the solid black lines. Owing to stronger systematics, we excluded the first ten integrations in the SZ reduction shown by the grey region at the beginning of each visit.

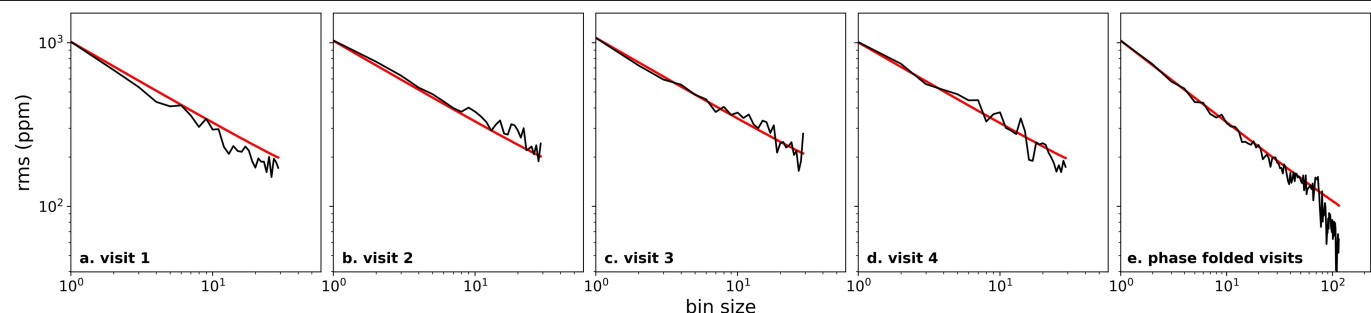

**Extended Data Fig. 4 | Allan deviation plots. a–d,** Allan deviation plots of the individual visits: rms of the best-fit residuals from data reduction SZ as a function of the number of data points per bin shown in black. **e,** The same but for the combined dataset. A bin size value of one corresponds to no binning. The red line shows the expected behaviour if the residuals are dominated by Gaussian noise. The absolute slope of this line is $1/\sqrt{bin\,size}$, following the inverse square root. The rms of our residuals closely follow this line, showing that our residuals are consistent with uncorrelated photon shot noise.

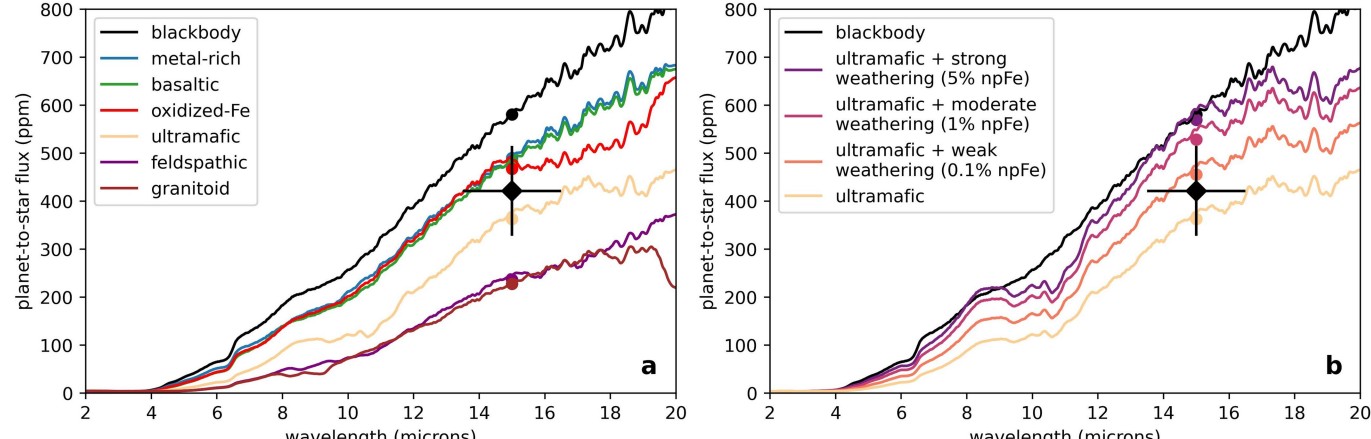

**Extended Data Fig. 5 | Measured eclipse depth compared with a suite of simulated bare-rock emission spectra. a**, Secondary eclipse spectra for various fresh surface compositions, assuming that TRAPPIST-1 c is a bare rock. High-albedo feldspathic and granitoid surfaces are cool and fit the data moderately poorly ($2\sigma$), as does a low-albedo and hot blackbody surface ($1.7\sigma$). **b**, Space weathering by means of formation of iron nanoparticles (npFe) lowers the albedo at short wavelengths, thereby increasing the surface's temperature and its secondary eclipse depth. An intermediate-albedo fresh ultramafic surface would fit the data well but the fit becomes marginal after taking into account the influence of strong space weathering ($1.6\sigma$, or about 90% confidence). The vertical error bar on our 15-µm measurement represents the $1\sigma$ uncertainty on the observed eclipse depth.

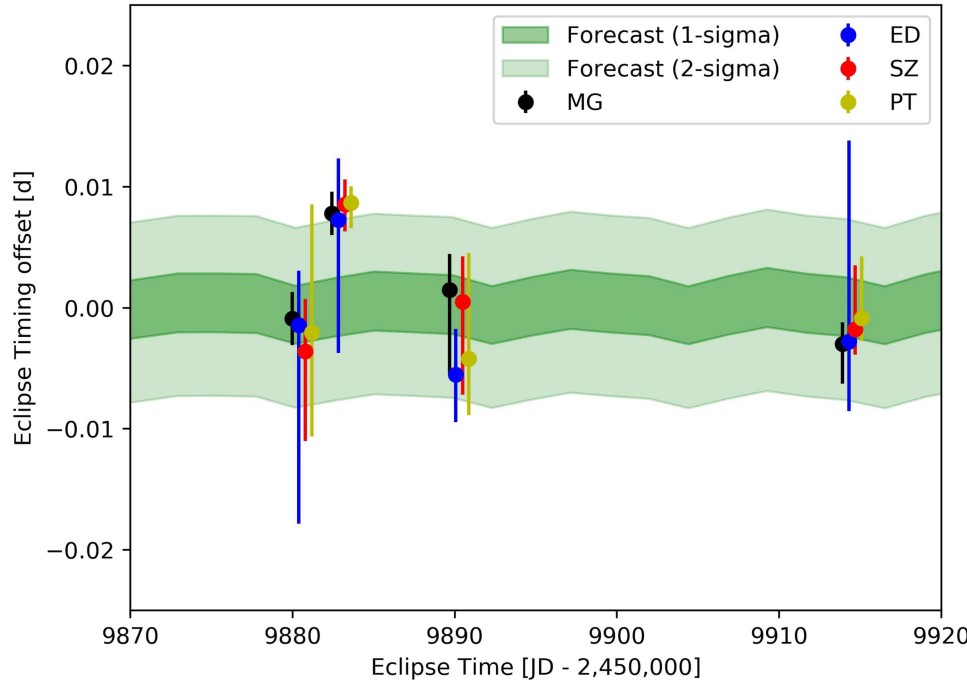

**Extended Data Fig. 6 | Measured eclipse times compared with the predicted eclipse times.** The points show the measured eclipse-timing offsets (defined as the time of eclipse minus the mean of the two adjacent transit times of planet c) from four different analyses. The error bars correspond to the 16th and 84th percentiles of the eclipse time posterior. The dark (light) green shaded region shows the $1\sigma$ ($2\sigma$) confidence intervals forecast from the transit-timing analysis from ref. 40.

**Extended Data Table 1 | Summary of the observations in JWST General Observer programme 2304**

|  | visit 1 | visit 2 | visit 3 | visit 4 |
|---|---|---|---|---|
| date | 27. Oct. 2022 | 30. Oct. 2022 | 6. Nov. 2022 | 30. Nov. 2022 |
| start time | 14:08:35 | 00:09:07 | 06:32:33 | 11:49:52 |
| end time | 17:21:29 | 03:21:23 | 09:44:49 | 15:02:47 |
| duration (hours) | 3.21 | 3.19 | 3.19 | 3.21 |
| Nint | 298 | 297 | 297 | 298 |
| Ngroups/int | 13 | 13 | 13 | 13 |
| stability rms x (pixel) | 0.0032 | 0.0040 | 0.0034 | 0.0031 |
| stability rms y (pixel) | 0.0059 | 0.0074 | 0.0062 | 0.0051 |

**Extended Data Table 2 | Details of the four different data reductions**

| Step/Parameter | SZ reduction | ED reduction | MG reduction | PT reduction |
|---|---|---|---|---|
| Stage 1 Run? | Yes | Yes | Yes | - |
| Jump correction | Jump rejection threshold of 7.0, 6.0, 7.0, 5.0 sigma | No jump correction | No jump correction | - |
| ramp weighting | default | uniform | uniform | - |
| Stage 2 Run? | Yes | Yes | Yes | - |
| photom step | skipped | skipped | skipped | - |
| Stage 3 notes | - | - | - | Used Calibration Level 2 data directly from MAST |
| centroid position determination method | 2D Gaussian fit to target | 2D Gaussian fit to target | 2D Gaussian fit to target | 2D Gaussian fit to target |
| target aperture shape | circle | circle | circle | circle |
| aperture radius | 4.0, 4.0, 4.0, 4.0 pixels around the centroid | 3.7, 4.0, 3.6, 3.8 pixels around the centroid | 3.6, 3.6, 3.5, 3.4 pixels around the centroid | 4.4, 4.1, 3.9, 3.5 pixels around the centroid |
| partial pixels treatment | pixels were supersampled using a bilinear interpolation | pixels were supersampled using a bilinear interpolation | used daophot/phot routine in IRAF | |
| background region shape | annulus around the centroid | annulus around the centroid | annulus around the centroid | annulus around the centroid |
| background aperture size | 25-41 for each visit | 20-35 for each visit | 20-35 for each visit | 30-45 for each visit |
| background subtraction method | subtracted the median calculated within the annulus from the whole frame | subtracted the median calculated within the annulus from the whole frame | Computation of the mode of the sky pixel distribution using the mean and median, after 3-sigma clipping of outliers. | mean of sigma-clipped pixel values within the annulus was subtracted from the whole frame (4-sigma clipping threshold) |
| Details of outlier rejection/time series clipping | No outliers removed with sigma clipping. first 10 integrations removed | sigma clipping set to 4, no exposure removed | 5-Sigma clipping with 20-min moving median. 5, 14, 6, and 4 integrations removed | No outliers removed with sigma clipping. First exposure removed from each visit |

**Extended Data Table 3 | Details of the four different data analyses**

| Step/Parameter | SZ reduction | ED reduction | MG reduction | PT reduction |
|---|---|---|---|---|
| Fitting method | `emcee` (MCMC) | `trafit` (MCMC-MH) | `trafit` (MCMC-MH) | `emcee` (MCMC) |
| Details for fitting method | 150,000 steps, 128 walkers, 30,000 as burn-in. Ran sampler for 80 times the autocorrelation length | 1 chain of 50,000 steps for error correction followed by 2 chains of 100,000 steps | 2 chains of 100,000 steps, with first 20% of chains as burn-in. Convergence checked with Gelman-Rubin statistical test. | 50,000 steps, 64 walkers, 5,000 as burn-in |
| total number of free parameters in the joint fit | 32 | 35 | 33 | 18 |
| number of free systematic parameters | 14 (in time) + 8 (decorr.) + 4 (uncertainty multiplier) | 14 (in time) + 11 (decorr.) | 12 (in time) + 5 (decorr.) | 11 (in time) |
| number of free astrophysical parameters | 6 (4 $f_p/f_*$ + $e$ + $\omega$) | 10 ($f_p/f_*$, $b$, 4 TTVs, $M_*$, $R_*$, $T_{\mathrm{eff}}$, [Fe/H])) | 16 ($f_p/f_*$, 7 TTVs, $\log \rho_*$, $\log M_*$, $T_{\mathrm{eff}}$, [Fe/H], $\cos i$, $(R_p/R_*)^2$, $\sqrt{e}\cos\omega$, $\sqrt{e}\sin\omega$) | 7 ($f_p/f_* + P_{orb}$ + $i$ + $a/R_*$ + $e$ + $\omega$ + $t_{sec}$) |
| rms of joint fit residuals | 1020 ppm | 961 ppm | 938 ppm | 1079 ppm |
| $f_p/f_*$ | $431^{+97}_{-96}$ ppm | $423^{+97}_{-95}$ ppm | $414 \pm 91$ ppm | $418^{+90}_{-91}$ ppm |

See Methods for more details on the individual fits. The uncertainties on the eclipse depth $f_p/f_*$ are 1σ.

## Extended Data Table 4 | Measured eclipse times

|    | visit 1 | visit 2 | visit 3 | visit 4 |
|----|---------|---------|---------|---------|
| SZ | $0.1872^{+0.0043}_{-0.0074}$ | $2.6209^{+0.0021}_{-0.0022}$ | $9.8782^{+0.0038}_{-0.0077}$ | $34.0940^{+0.0053}_{-0.0021}$ |
| ED | $0.1894^{+0.0452}_{-0.0164}$ | $2.6197^{+0.0051}_{-0.0110}$ | $9.8722^{+0.0038}_{-0.0040}$ | $34.0930^{+0.0166}_{-0.0057}$ |
| MG | $0.1899 \pm 0.0022$ | $2.6202 \pm 0.0018$ | $9.8792^{+0.0033}_{-0.0069}$ | $34.0928^{+0.0018}_{-0.0030}$ |
| PT | $0.1887^{+0.0106}_{-0.0086}$ | $2.6211^{+0.0014}_{-0.0021}$ | $9.8735^{+0.0087}_{-0.0047}$ | $34.0949^{+0.0051}_{-0.0019}$ |

Eclipse times (in $BJD_{TDB}$) determined by the four different reductions by fitting an eclipse model to each visit. An offset of 2459880.0 days was subtracted from each of the values in the table.