## [Peer Review File · Nature]

Manuscript Title: No thick carbon dioxide atmosphere on the rocky exoplanet TRAPPIST-1 c

Reviewer Comments & Author Rebuttals

Reviewer Reports on the Initial Version:

Referees' comments:

Referee #1 (Remarks to the Author):

No thick carbon dioxide atmosphere on the rocky exoplanet TRAPPIST-1 c

Zueba et al.

Summary:

The authors present constraints on the thermal emission of TRAPPIST-1c at 15 microns. This single photometric point from JWST provides sufficient information to rule out a thick, CO₂-rich atmosphere; however, atmospheres of <10 bars (including tenuous and no atmospheres) are still possible. The data reduction methods are robust and follow best practices. Multiple analyses produce consistent results and the interpretation appears reliable. The manuscript is well written but, as indicated in my comments below, there are opportunities to improve the clarity of the information presented.

Major Concerns:

1. Nature recently published the non-detection of an atmosphere for TRAPPIST-1b using the same MIRI filter (Greene et al., 2023). Given that this is now the second non-detection for a planet in the TRAPPIST-1 system, it's difficult to see what makes this paper novel. I sympathize with the authors on this concern because, as I understand it, this paper was submitted before results from the TRAPPIST-1b paper were announced.
2. In Figure 4, I'm struggling to follow the argument that the initial water abundance cannot exceed 10 Earth oceans. Assuming the final surface pressure is 100 bars (the stated upper limit) and the XUV saturation fraction is 1E-3, then the initial water abundance looks like it can be anywhere from 10-100 Earth oceans. Perhaps the color bar and plot need to be extended to include final surface pressures >100 bars?
3. How robust is the assumption that the XUV saturation fraction is 1E-3? In other words, what's the uncertainty on this value? I searched through Chadney et al. (2014), but could not find an answer. My expectation is the uncertainty is about an order of magnitude, which means the predicted range in initial water abundance should be much larger (e.g., 1 - 100 Earth oceans). This would then broaden the inferred "volatile-poor" formation history. Ultimately, my concern is that the authors' conclusion on this matter is not robust.

Minor Comments:

Line 86: I suggest adding a reference to the recently-published TRAPPIST-1b paper.

Line 128/Figure 1 caption: The text reports that Figure 1 shows one of the four light curves; however, the caption and reported depth/uncertainty suggest the figure is showing an average of the four eclipses. I suggest presenting the combined light curve, if you aren't already.

Lines 138-140: Now that the TRAPPIST-1b paper has been published, these sentences need to be updated. Please also update Figure 8.

Line 142: To save the reader some time, please include the albedo values for Mercury and Venus in this manuscript.

Line 173: I believe this should be the "observed *bulk* density"

Line 222: As a planetary scientist who comfortably breathes in a 1-bar atmosphere, I object to a 10-bar atmosphere being described as "thin." Please rephrase to provide an unbiased account.

Tables 2+4: It would be helpful if the reduction order in Tables 2+4 matched the description order in the Data Reduction and Data Analysis sections. On several occasions, I found myself referencing the wrong column.

Figure 6: Any guesses as to why the background was so much higher for Visit 4?

Line 332: Presumably you excluded the first integration, not frame.

Line 423: Why is the final ED eclipse depth (423 ppm) not particularly close to the average eclipse depth from the individual visits (mean(445,418,474,459) = 449 ppm)?

Line 519: Did you use only the out-of-eclipse data when estimating the stellar flux density?

Line 536: This discrepancy in the brightness temperature of the star is disconcerting. Could this be related to inaccuracies in the actual gain value? What gain value did MG apply here? How does this temperature compare to estimates using Spitzer data?

Trivial Matters:

Fig 2 caption: "suit" should be "suite"

Table 2, Details of outlier rejection: Did you mean integrations removed, not exposures removed?

Figure 7 caption: "...dominated by *uncorrelated* noise."

Line 665: "condensate" should be "condense"

Referee #2 (Remarks to the Author):

The manuscript "No thick carbon dioxide atmosphere on the rocky exoplanet TRAPPIST-1 c" by S. Zieba et al. presents the first secondary eclipse observations of the exoplanet TRAPPIST-1 c. TRAPPIST-1 c is similar in size, mass, and irradiation to Venus. The key scientific finding is that unlike Venus, the planet does not have a high-pressure atmosphere predominantly composed of CO₂. This is the first work to characterize an exoplanet that is Venus-like or reasonably similar to Earth, and its comparison to models shows that TRAPPIST-1 c likely formed with relatively small amounts of water. Overall, this manuscript is a significant milestone in characterizing terrestrial exoplanets. Together with the very recent similar study of TRAPPIST-1 b (Greene et al. 2023), its results will propel modeling the formation and evolution of M-star rocky planets as well as guide future observations of these systems.

The manuscript presents a careful and thorough analysis of the data and shows that four independent analyses arrive at similar results. This is reassuring and important given the newness of JWST mid-infrared observations and the potential high impact of the work. The authors have also interpreted the results as thoroughly as justified by the somewhat limited data; they have compared the observations to existing models and have also developed new models of bare rock surfaces and O₂-dominated atmospheres. The manuscript thoughtfully presents and compares a range of surface and atmospheric models to the data and makes reasonable conclusions that validly consider the data uncertainties. It is also referenced thoroughly and generally written clearly with a few exceptions noted below.

I strongly recommend the paper for publication in Nature after the authors consider and address the following minor comments and suggestions:

Main section:

Line 185: It would be helpful to say "... for both cases, the emitting layer and cloud deck lie at similar pressures" as explained on line 619 in Methods.

Lines 193 - 205 provide a good summary of what rock compositions and weathering are consistent with the observations. The text should also reference Figure 3 since that shows a presumably unweathered ultramafic rock simulated spectrum.

Figure 3: The caption should state that the ultramafic rock spectrum was made with no space weathering, and that may not be realistic for TRAPPIST-1 c (per lines 567 - 575).

Lines 207 - 220: It is not clear whether this modeling of initial water content included the impact of an extended luminous pre-main-sequence phase for TRAPPIST-1 (refs 7, 66). The text should be more clear on this, and the implications (higher initial water content?) should be discussed if this was not considered.

Figure 2: It is likely obvious to many readers, but it would be helpful to state in the caption that the difference between each model and the observation is given as a numerical value in units of sigma in each grid point. This is implied in the scale color bar but perhaps confusing because the figure shows a greater range of values than in the scale bar.

Figure 4: The figure does not clearly show how the observations constrain the initial water abundance. It would be helpful to add the observed surface pressure upper limit to the figure to help convey how this constrains the initial water abundance. The caption does not explain the meaning of the white numbers overlayed in the bottom half of the figure - are these the allowed range of atmospheric pressures in log bars? The negative values are also difficult to read due to poor contrast with the colors in the figure. The authors should also consider moving this figure to Methods after these revisions if space is short; I do not think it is as impactful as the other three figures in the Main paper.

Methods:

Line 235: The publicly-available APT file for JWST program 2304 shows that the data were acquired with the FASTR1 readout mode, not FAST. It would be worthwhile to correct this because both modes may be offered in the future, and they have different integration duty cycles and noise characteristics.

Table 2 is very helpful. It would be more readable if the longer row names were shortened to allow some vertical space between them. For example, "size of aperture" could be changed to "aperture radius" which would be shorter and more descriptive as well. Similar changes should be made to most other row headings as well.

Figure 6: Does this figure apply specifically to the SZ reductions, or is it generally applicable to all? This should be stated in the caption. The units of the top 2 rows do not seem to make sense, and the term "raw flux" is not explained. The figure implies that the raw flux is $\sim 3.5E6$ electrons per second, about 100 x the background. That ratio seems excessively high, and both numbers seem impossibly high if they are per-pixel values. I think that it would make the most sense to present them as per-pixel values in their respective apertures, and this should be explained in the caption.

Line 293: The text says that the TRAPPIST-1 observations had 17 frames per integration, but line 236 states 13 groups per integration which is consistent with the APT file. The value should be corrected here, and it would also be helpful to use consistent terminology throughout (frames vs. groups).

Line 332: Is it correct that only the first frame of each visit was discarded, or was the first

integration discarded? Lines 325 - 327 suggest that the author of this section used the term "frame" instead of "integration" (standard JWST convention) to refer to each set of groups collected between detector resets. This is also confusing because the STScI pipeline has a "first frame correction" step for MIRI data that I did not see addressed explicitly in the Methods section.

Table 4 is also very helpful. Together with Table 2, it clearly shows how the data were reduced and processed by the different authors. Both should be retained in the article.

Figure 7: The caption should indicate that these Allan variance plots were computed as part of the SZ data analysis. It would be helpful to note here or in the text whether the other analyses showed similar noise properties if they were computed. There is no need to compute them for the other analyses if this has not been done.

Figure 8 is interesting and worthwhile, but it is not clear why this appears in Methods. It is only referenced in the Main text (Line 138), and I could not find if or where it was described in the Methods section. It seems out of place here. Also, are the error bars 1 sigma?

Lines 517 - 537: This appears to be the only analysis that computes the stellar brightness and then the planet's brightness and brightness temperature. This analysis is also outside the scope of the analysis stages presented in Table 4. Therefore this section should be given its own subheading in Methods to make clear that it describes the computation of the stellar and planet fluxes and brightness temperatures. Information on what data products were used to compute the brightnesses and the calibration versions should also be added. This is relevant because the JWST calibration continues to evolve, and a better future calibration could change the measured fluxes and brightness temperatures.

Figure 9 illustrates the discussion in the Bare Rock section well. That section, lines 545 - 575, does not reference this figure but it should. If there were space, it would be good to include this figure in the Main section of the manuscript, where it would complement what Figure 2 shows for atmospheric models. The authors might consider moving Figure 9 to the Main section and Figure 4 to Methods to conserve space.

Figure 10: I believe that the caption should be edited to say that the points (not error bars) show the measured eclipse timing offsets. The value of the error bars should be stated (1 sigma?).

Author Rebuttals to Initial Comments:

Referee #1

Major Concerns:

1. Nature recently published the non-detection of an atmosphere for TRAPPIST-1b using the same MIRI filter (Greene et al., 2023). Given that this is now the second non-detection for a planet in the TRAPPIST-1 system, it's difficult to see what makes this paper novel. I sympathize with the authors on this concern because, as I understand it, this paper was submitted before results from the TRAPPIST-1b paper were announced.

We did now add the Greene et al., 2023 reference in our paper.

While it's true that both planets are in the TRAPPIST-1 system and the type of observation is the same, we think there are several important differences between the results. For example: the TRAPPIST-1 b measurement is more consistent with a null-albedo bare-rock. Our observations of TRAPPIST-1 c are more consistent with a thin atmosphere and/or a bare-rock with a non-zero albedo surface. Furthermore, TRAPPIST-1 c is colder than b (2x Earth insolation, as opposed to 4x for planet b). It is in fact the most temperate Earth-sized planet with a thermal emission detection.

Our Extended Figure 1 show that TRAPPIST-1 c is now:

- 1) the smallest rocky transiting exoplanet with observed thermal emission (T1b and T1c are nearly the same size but observations have shown a slightly smaller radius for T1c: $1.116 +0.014 -0.012 R_e$ for T1b vs $1.097 +0.014 -0.012 R_e$ for T1c).
- 2) the coldest transiting exoplanet with a measured emission in the infrared: T1c's brightness temperature is approximately 120 Kelvin lower than T1b's brightness temperature at 15 microns (380 ± 31 K for T1c vs. 503 ± 27 K for T1b).

2. In Figure 4, I'm struggling to follow the argument that the initial water abundance cannot exceed 10 Earth oceans. Assuming the final surface pressure is 100 bars (the stated upper limit) and the XUV saturation fraction is $1E-3$, then the initial water abundance looks like it can be anywhere from 10-100 Earth oceans. Perhaps the color bar and plot need to be extended to include final surface pressures >100 bars?

The contour plot shows the final atmospheric pressure for the given initial water abundances and XUV saturation fractions. For an upper permitted atmospheric pressure of 100 bars, only the space below the 100 bar (= 2) contour with lower atmospheric pressures matches the observations. All of the space colored black produces atmospheres with pressures greater than 100 bars and is therefore not allowed by the observations.

We added a shaded area in the figure, so that it is clearer which parameter space is the "excluded regime".

3. How robust is the assumption that the XUV saturation fraction is $1E-3$? In other words, what's the uncertainty on this value? I searched through Chadney et al. (2014), but could not find an answer. My expectation is the uncertainty is about an order of magnitude, which means the predicted range in initial water abundance should be much larger (e.g., 1 - 100 Earth oceans). This would then broaden the inferred "volatile-poor" formation history. Ultimately, my concern is that the authors' conclusion on this matter is not robust.

How robust is the assumption that the XUV saturation fraction is $1E-3$? Wright et al. (2018) MNRAS 479, 2351 (<https://ui.adsabs.harvard.edu/abs/2018MNRAS.479.2351W/abstract>) give Xray saturation fractions for a sample of young M dwarfs. The saturation fraction at early times is best fit by $1e-3$, but does vary between about $5e-4$ to $5e-3$. Fleming et al. (2020) (<https://ui.adsabs.harvard.edu/abs/2020ApJ...891..155F/abstract>) used stellar models to estimate the XUV saturation fraction for Trappist-1 specifically, and found a best fit of $10^{-3.03}$ (+0.23/-0.12). We added references to these two papers in the Method section where we discuss the atmospheric escape.

We now also show the errorbars on this XUV saturation fraction measurement in Figure 4 on reference Fleming et al, 2020 there too. We furthermore added errorbars to the 4 or 10 Earth oceans measurements based on the uncertainties estimated by Fleming et al, 2020. So, we get 4 (+1.3/-0.8) oceans for 10 bars and for 100 bars, I get 9.5 (+7.5/-2.3) oceans.

Minor Comments:

Line 86: I suggest adding a reference to the recently-published TRAPPIST-1b paper.

Thank you. We added a reference to the recently published TRAPPIST-1 b paper at the beginning of the main text and in the abstract.

Line 128/Figure 1 caption: The text reports that Figure 1 shows one of the four light curves; however, the caption and reported depth/uncertainty suggest the figure is showing an average of the four eclipses. I suggest presenting the combined light curve, if you aren't already.

We make it clear now in the caption that the figure is showing the light curve and fit based on the SZ reduction.

The results from all 4 reductions were almost identical. So we decided against showing the light curves and fits from each reduction because they are so similar that one wouldn't even notice a difference between the figures.

Lines 138-140: Now that the TRAPPIST-1b paper has been published, these sentences needs to be updated. Please also update Figure 8.

Thank you. We now also mention the observed brightness temperature for TRAPPIST-1 b and write that measured temperatures for T1b and T1c are more than 500 lower than the previous "recond holder" LHS3884b

at 1,040K. We also did update figure 8 (which is now Extended Figure 1) to include the recently published results by Greene et al., 2023.

Line 142: To save the reader some time, please include the albedo values for Mercury and Venus in this manuscript.

Thank you. We added the Bond albedos.

Line 173: I believe this should be the "observed *bulk* density"

Thank you. We added "bulk".

Line 222: As a planetary scientist who comfortably breathes in a 1-bar atmosphere, I object to a 10-bar atmosphere being described as "thin." Please rephrase to provide an unbiased account.

Thank you. We did change the phrasing to "moderately thin".

Tables 2+4: It would be helpful if the reduction order in Tables 2+4 matched the description order in the Data Reduction and Data Analysis sections. On several occasions, I found myself referencing the wrong column.

Thank you. We agree. We now have a consistent order (SZ,ED, MH, PT) in the Methods text and tables.

Figure 6: Any guesses as to why the background was so much higher for Visit 4?

The background flux depends on several factors like the orientation of the telescope, the angle of the telescope with respect to the zodiacal dust, and the temperature of the spacecraft. These might have caused slightly different background fluxes.

The observations were taken on: 27. Oct. 2022, 30. Oct. 2022, 6. Nov. 2022, and 30. Nov. 2022. So, the 4th visit was therefore taken several weeks after the previous ones. During this time, JWST covered some distance in our solar system. This changed location might have ultimately influenced the background flux.

The JWST docs state: "MIRI imaging sensitivity is [...] astronomical background limited at wavelengths <15 μm and telescope background (primary mirror and sunshield) limited at wavelengths >15 μm ". Ultimately, because we cannot pinpoint the dominating cause of the change in background and it does not significantly affect our data precision, we did not include any speculation on the background flux in the paper.

Line 332: Presumably you excluded the first integration, not frame.

Thank you. That's correct. We changed "frame" to "integration".

Line 423: Why is the final ED eclipse depth (423 ppm) not particularly close to the average eclipse depth from the individual visits ($\text{mean}(445,418,474,459) = 449 \text{ ppm}$)?

The joint fit is an independent analysis with different priors. For the individual analyses t_0 and b (the impact parameter) were jump parameters whereas for the joint fit only the TTVs were jump parameter for the planet. In that regard, the posterior distributions of all four individual visits are consistent with the one from the joint fit but the mean value from the joint fit's posterior is not necessarily equal to the mean of the depths measured for each individual visit. For illustration we have re-ran all four individual analyses as well as the joint one to compare the posteriors, see figure attached.

Line 519: Did you use only the out-of-eclipse data when estimating the stellar flux density?

No, but this does not matter as the eclipse depth is only 0.04% while the absolute photometric error is estimated to be 3% for MIRI.

Line 536: This discrepancy in the brightness temperature of the star is disconcerting. Could this be related to inaccuracies in the actual gain value? What gain value did MG apply here? How does this temperature compare to estimates using Spitzer data?

We were a bit surprised too, but actually this discrepancy is predicted by atmospheric models. The reason that T_b at 15 microns is that the SED of such a cold star is not a blackbody. CO and H₂O are expected to absorb at 15 microns in the stellar photosphere. The counts in the MIRI images are already in MJy/sr, so assuming that the MIRI calibration has been well done (and the MIRI team claims that it was well done with a typical precision of 3%), these absolute fluxes are reliable.

So this discrepancy is expected. Furthermore, the atmospheres of stars at this temperature are strongly non-gray, so that the temperature keeps decreasing with pressure. This causes the $\tau \sim 1$ depth of the star to be at lower pressure/temperature, so that the brightness temperature at 15 microns is substantially less than the effective temperature.

MG used a value of 3.1 e⁻/ADU for the gain which we mentioned in the data reduction subsection of MG.

We did not further compare our measured stellar brightness to measurements by Spitzer. We believe this is outside of the scope of the paper. But with the precise photometry measurements of the star by JWST (MIRI observed the star at 15 microns and at 12.8 microns, the latter being a GTO program) and the previous observations by other observatories, we are sure that a follow-up study on the star's SED will follow.

Trivial Matters:

Fig 2 caption: "suit" should be "suite"

Yes, thank you. Done

Table 2, Details of outlier rejection: Did you mean integrations removed, not exposures removed?

Yes, thank you. We corrected this.

Figure 7 caption: "...dominated by *uncorrelated* noise."

Thank you. We rephrased this to make it clearer:

Previously: "The rms of our residuals closely follow this line, showing that we are not dominated by correlated noise."

Now: "The rms of our residuals closely follow this line, showing that our residuals are consistent with uncorrelated photon shot noise."

Line 665: "condensate" should be "condense"

Thank you. We corrected this.

Referee #2

Main section:

Line 185: It would be helpful to say "... for both cases, the emitting layer and cloud deck lie at similar pressures" as explained on line 619 in Methods.

Thank you. We agree that this improves the clarity and we therefore adjusted the text as suggested.

Lines 193 - 205 provide a good summary of what rock compositions and weathering are consistent with the observations. The text should also reference Figure 3 since that shows a presumably unweathered ultramafic rock simulated spectrum.

Thank you. We agree that referencing Fig 3 is useful for the reader. We now reference Figure 3 in the text: "... (see Fig. 3 for an unweathered ultramafic surface and Fig. 9 for all surfaces we considered)"

Figure 3: The caption should state that the ultramafic rock spectrum was made with no space weathering, and that may not be realistic for TRAPPIST-1 c (per lines 567 - 575).

We now state that the ultramafic spectrum is without weathering. We also refer the reader to the main text if they want to read more on weathering. As we don't really know the age of the surface, we don't precisely know the amount of weathering the planet experienced. We picked an unweathered surface for this plot but add further discussion about this topic in the Lines 193 - 205 and the Methods bare-rock section.

Lines 207 - 220: It is not clear whether this modeling of initial water content included the impact of an extended luminous pre-main-sequence phase for TRAPPIST-1 (refs 7, 66). The text should be more clear on this, and the implications (higher initial water content?) should be discussed if this was not considered.

Yes, the extended pre-main sequence was considered. We utilized the stellar evolution models of Baraffe et al. (2015) for a 0.09 Msun star to approximate the pre-main sequence evolution of TRAPPIST-1.

We mentioned this remark in the Methods section of the manuscript.

Figure 2: It is likely obvious to many readers, but it would be helpful to state in the caption that the difference between each model and the observation is given as a numerical value in units of sigma in each grid point. This is implied in the scale color bar but perhaps confusing because the figure shows a greater range of values than in the scale bar.

Thank you. We agree that this might improve the understandability of the figure. We added an explanation to the figure caption.

Figure 4: The figure does not clearly show how the observations constrain the initial water abundance. It would be helpful to add the observed surface pressure upper limit to the figure to help convey how this constrains the initial water abundance. The caption does not explain the meaning of the white numbers overlaid in the bottom half of the figure - are these the allowed range of atmospheric pressures in log bars? The negative values are also difficult to read due to poor contrast with the colors in the figure. The authors should also consider moving this figure to Methods after these revisions if space is short; I do not think it is as impactful as the other three figures in the Main paper.

Thank you. We tried to improve the clarity of this figure and caption by considering the changes suggested by the referee:

- We added a hatched area showing the excluded area, i.e., where the pressure is greater than 100 bars.
- We added an explanation on the white numbers to the caption saying: "The white numbers are the contour values for the logarithm of the atmospheric pressure in bars."
- We also made these numbers better readable by making them bold and decreasing their opacity.

Because we mention our conclusions based on the TRAPPIST-1 c water content and the ramifications of this for the whole TRAPPIST-1 system in the abstract, we think this figure plays an important role for the understanding of our argumentation. We therefore retained Figure 4 in the main text.

Methods:

Line 235: The publicly-available APT file for JWST program 2304 shows that the data were acquired with the FASTR1 readout mode, not FAST. It would be worthwhile to correct this because both modes may be offered in the future, and they have different integration duty cycles and noise characteristics.

Thank you, this is totally correct. We corrected FAST to FASTR1.

Table 2 is very helpful. It would be more readable if the longer row names were shortened to allow some vertical space between them. For example, "size of aperture" could be changed to "aperture radius" which would be shorter and more descriptive as well. Similar changes should be made to most other row headings as well.

Thank you. We worked on making the row headings more compact. We also increased the line spacing in table 2 and 4 to improve readability.

Figure 6: Does this figure apply specifically to the SZ reductions, or is it generally applicable to all? This should be stated in the caption. The units of the top 2 rows do not seem to make sense, and the term "raw flux" is not explained. The figure implies that the raw flux is $\sim 3.5E6$ electrons per second, about 100 x the background. That ratio seems excessively high, and both numbers seem impossibly high if they are per-pixel values. I think that it would make the most sense to present them as per-pixel values in their respective apertures, and this should be explained in the caption.

Thank you. We mention in the caption that this diagnostic plot is referring to the SZ reduction.

We converted the flux values into flux-per-pixel values. As an example, for visit 1, the flux within the aperture is approximately 69200 electrons per integration per pixel and the background flux is 30500 electrons per integration per pixel. This means that the background is making up $\sim 44\%$ ($=30500/69200$) of the flux within the aperture. This is in agreement with Greene et al., 2023 who also find a similar value and conclude that background subtraction is important for MIRI photometry (taken from the beginning of the Method section from

Greene et al., 2023): “This background level corresponds to approximately 45% of the total flux within a five-pixel radius photometric aperture centered on the TRAPPIST-1 star.”

Line 293: The text says that the TRAPPIST-1 observations had 17 frames per integration, but line 236 states 13 groups per integration which is consistent with the APT file. The value should be corrected here, and it would also be helpful to use consistent terminology throughout (frames vs. groups).

We thank the referee for pointing this out. There were in fact 13 groups per integration. It was only wrongly stated in this part of the manuscript. We also adopted the “groups” terminology throughout the paper and removed any instance of “frames” in the subsection “Data Reduction ED”.

Line 332: Is it correct that only the first frame of each visit was discarded, or was the first integration discarded? Lines 325 - 327 suggest that the author of this section used the term "frame" instead of "integration" (standard JWST convention) to refer to each set of groups collected between detector resets. This is also confusing because the STScI pipeline has a "first frame correction" step for MIRI data that I did not see addressed explicitly in the Methods section.

Thank you for pointing out that the word “frame” was misleading and not correctly used here. This particular reduction (Data Reduction PT) did remove the first *integration*. We therefore corrected all instances of “frames” here with “integrations”.

We did not explicitly address the “first” or “last frame correction” because we used the default settings of the JWST calibration pipeline for MIRI TSO data (except from using certain jump rejection threshold values or skipping the photom step which we mentioned in the text). Because we used the default settings (as mentioned in the text), we skipped the first and last frame correction steps.

Table 4 is also very helpful. Together with Table 2, it clearly shows how the data were reduced and processed by the different authors. Both should be retained in the article.

We agree that Table 2 and 4 are helpful and that they provide the reader with a data reduction and data fitting overview for all 4 reductions. We however kept these tables in the Method section because they are rather technical and feel that they are more suited as Method section content.

We did however reference these two tables in the Main text now and say that the Extended Data Tables contain tables on the data reduction and data fitting.

We hope this is sufficient for the referee.

Figure 7: The caption should indicate that these Allan variance plots were computed as part of the SZ data analysis. It would be helpful to note here or in the text whether the other analyses showed similar

noise properties if they were computed. There is no need to compute them for the other analyses if this has not been done.

Thank you. These Allan deviation plots are indeed from the SZ reduction and we now mention that in the caption. We also did these plots for the MG reduction and they also show this nice behavior as seen in Fig. 7. We did not include the Allan deviation plots from the MG reduction or any other reduction because we did not think it would really contribute any additional information.

We did add two sentences to the MG analysis text stating that we did create Allan deviation plots for each visit and that we basically see the same behavior as in Fig. 7.

Figure 8 is interesting and worthwhile, but it is not clear why this appears in Methods. It is only referenced in the Main text (Line 138), and I could not find if or where it was described in the Methods section. It seems out of place here. Also, are the error bars 1 sigma?

Thank you. The placement of the figure was in fact not following the guidelines. It is the first Methods-figure mentioned in the main text. It therefore has to appear as the first figure in the Methods. We changed this, to follow the journal guidelines.

The error bars are one sigma. Thank you. We added that to the caption.

Lines 517 - 537: This appears to be the only analysis that computes the stellar brightness and then the planet's brightness and brightness temperature. This analysis is also outside the scope of the analysis stages presented in Table 4. Therefore this section should be given its own subheading in Methods to make clear that it describes the computation of the stellar and planet fluxes and brightness temperatures. Information on what data products were used to compute the brightnesses and the calibration versions should also be added. This is relevant because the JWST calibration continues to evolve, and a better future calibration could change the measured fluxes and brightness temperatures.

Thank you, we moved the calculation of the stellar and planet's brightness into its own subsection. We also added more information on the data product and calibration used. We now note at the beginning of this new subsection the following: "The following analysis was based on stage 0 (.uncal) data products pre-processed by the JWST data processing software version number 2022_3b, and calibrated with \texttt{Eureka!} as described above in section "Data Reduction MG"."

Figure 9 illustrates the discussion in the Bare Rock section well. That section, lines 545 - 575, does not reference this figure but it should. If there were space, it would be good to include this figure in the Main section of the manuscript, where it would complement what Figure 2 shows for atmospheric models. The authors might consider moving Figure 9 to the Main section and Figure 4 to Methods to conserve space.

Done. We are now referencing the figure in the bare rock section. We still think Figure 4 is a little bit more significant for the interpretations of the results and Figure 9 (the Figure showing the bare rock models) is pretty detailed. So we decided to keep Fig 4 in the main text.

Figure 10: I believe that the caption should be edited to say that the points (not error bars) show the measured eclipse timing offsets. The value of the error bars should be stated (1 sigma?).

Thanks you. Done. We wrote that the errorbars are showing the 16th and 84th percentile of the eclipse time posterior.